# The Role of Omega-3 Polyunsaturated Fatty Acids from Different Sources in Bone Development

**DOI:** 10.3390/nu12113494

**Published:** 2020-11-13

**Authors:** Reut Rozner, Janna Vernikov, Shelley Griess-Fishheimer, Tamar Travinsky, Svetlana Penn, Betty Schwartz, Ronit Mesilati-Stahy, Nurit Argov-Argaman, Ron Shahar, Efrat Monsonego-Ornan

**Affiliations:** 1The Robert H. Smith Faculty of Agriculture, Food and Environment, Institute of Biochemistry and Nutrition, The Hebrew University of Jerusalem, Rehovot 7610001, Israel; reut.rozner@mail.huji.ac.il (R.R.); janna444@gmail.com (J.V.); shelley.gg@gmail.com (S.G.-F.); tamarmura@gmail.com (T.T.); s9473net@yahoo.com (S.P.); betty.schwartz@mail.huji.ac.il (B.S.); 2Animal Science, The Robert H. Smith Faculty of Agriculture, Food and Environment, The Hebrew University of Jerusalem, Rehovot 7610001, Israel; ronit_mesilati@walla.co.il (R.M.-S.); argov.nurit@mail.huji.ac.il (N.A.-A.); 3The Robert H. Smith Faculty of Agriculture, Food and Environment, Koret School of Veterinary Medicine, The Hebrew University of Jerusalem, Rehovot 7610001, Israel; ron.shahar1@mail.huji.ac.il

**Keywords:** n-3 fatty acids, n-6 fatty acids, growth plate, bone mechanics, RNA-sequencing

## Abstract

N-3 polyunsaturated fatty acids (PUFAs) are essential nutrients that must be obtained from the diet. We have previously showed that endogenous n-3 PUFAs contribute to skeletal development and bone quality in fat-1 mice. Unlike other mammals, these transgenic mice, carry the n-3 desaturase gene and thus can convert n-6 to n-3 PUFAs endogenously. Since this model does not mimic dietary exposure to n-3 PUFAs, diets rich in fish and flaxseed oils were used to further elucidate the role of n-3 PUFAs in bone development. Our investigation reveals that dietary n-3 PUFAs decrease fat accumulation in the liver, lower serum fat levels, and alter fatty acid (FA) content in liver and serum. Bone analyses show that n-3 PUFAs improve mechanical properties, which were measured using a three-point bending test, but exert complex effects on bone structure that vary according to its source. In a micro-CT analysis, we found that the flaxseed oil diet improves trabecular bone micro-architecture, whereas the fish oil diet promotes higher bone mineral density (BMD) with no effect on trabecular bone. The transcriptome characterization of bone by RNA-seq identified regulatory mechanisms of n-3 PUFAs via modulation of the cell cycle and peripheral circadian rhythm genes. These results extend our knowledge and provide insights into the molecular mechanisms of bone remodeling regulation induced by different sources of dietary n-3 PUFAs.

## 1. Introduction

The skeleton is a unique system that performs mechanical, storing, metabolic, and protective functions. Despite its inert appearance, bone is a highly dynamic organ undergoing constant remodeling in order to maintain optimum mechanical functions. Bone tissue is continuously degraded by bone resorbing cells—the osteoclasts—which are replaced by new bone-forming cells called osteoblasts [1]. The development and differentiation of these two distinct cells are tightly regulated by numerous endogenous substances, including growth factors, hormones, cytokines, and neurotransmitters. Although genetics contribute considerably to peak bone mass (PBM), an individual’s full genetic potential can be achieved through proper nutrition and exercise [2].

Fractures are a major public health concern in females as well as males. Bone density, one of the key predictors of osteoporotic fractures, is the result of the amount of bone gained in early life and subsequent bone loss [3]. Existing treatment for osteoporosis includes mostly pharmacological approaches to inhibit bone resorption [4]; however, strategies that can improve bone health and functional performance are likely to provide the greatest protection against fractures. A preferable tactic to avoid osteoporosis is focusing on prevention strategies such as nutrition during the growth phase in order to optimize PBM [5].

Evidence presented over recent years has shown that n-3 polyunsaturated fatty acids (PUFAs) are beneficial for bone health. Högström et al. reported that n-3 PUFAs were positively associated with higher bone mineral density (BMD) in healthy young men [6]. Lukas et al. showed that rats that were fed a high-fat diet rich in n-3 PUFAs had higher tibial BMD and improved trabecular bone micro-architecture, respectively [7]. Linoleic acid (LA 18:2 cis-9.12) and alpha linolenic acid (ALA 18:3 cis-9.12.15) belong to the n-6 and n-3 series of PUFAs, respectively, and serve as the precursors for all other n-6 and n-3 PUFA derivatives. Since mammals lack the enzyme n-3 desaturase [8], which is required for producing LA from oleic acid (18:1 cis-9) and ALA from LA, these fatty acids (FAs) are considered essential FAs (EFAs) and must be derived from the diet [9]. After digestion, EFAs can be converted to their higher unsaturated PUFA derivatives: arachidonic acid (AA) from LA and eicosapentaenoic (EPA) and docosahexaenoic acids (DHA) from ALA through desaturation and elongation [3].

The cell’s membrane structure is affected by its FA content. The long chain (LC) PUFAs of both the n-6 and n-3 series contribute to membrane fluidity and serve as precursors of various lipid-mediated signaling molecules called eicosanoids [3,10]. Furthermore, their biological activities include regulation of intracellular signaling pathways, transcription factor activity, and gene expression. Through these effects, LC-PUFAs modify cellular functions and influence health and disease risk [11]. Several mechanisms have been suggested as mediators of the effects of dietary fats on bone, including alterations in calcium absorption, prostaglandin synthesis, down-regulation of osteoclastogenesis or up-regulation of osteoblastogenesis [12].

Industrialization over recent decades has shifted the ratio in Western diets between n-6 PUFAs and n-3 PUFAs to 15–20:1, whereas the recommended ratio is approximately 4:1 [13]. LA is the most prevalent n-6 PUFA in the diet and exists in various plant oils such as corn oil, sunflower oil, and soybean oil. ALA is found in several plant sources such as flaxseed oil and walnuts, whereas its derivatives, EPA and DHA, are found mainly in marine oils [14].

We have previously showed a positive effect of n-3 PUFAs on bone development and quality in fat-1 transgenic mice (expressing the c-elegans n-3 desaturase). Endogenous n-3 PUFAs that were produced during the growth period resulted in an improved trabecular and cortical bone micro-architecture and stronger bone compared with control mice [15]. However, this transgenic model does not mimic human dietary exposure to n-3 PUFAs. In the present study, therefore, diets rich in n-3 PUFAs were used to further elucidate the role of different sources of n-3 PUFAs in bone development. In addition, to examine the molecular mechanism of bone formation induced by n-3 PUFAs, RNA-seq analysis of whole bones was performed.

## 2. Materials and Methods

### 2.1. Animals

Forty-eight female C57BL6 mice after weaning were purchased from Harlen Laboratories (Rehovot, Israel) and housed in environmentally controlled conditions. All procedures were approved by the Hebrew University Animal Care Committee (permit number AG-17-15179-2). The mice were fed a normal chow diet for the first 4 days and were then randomly divided into 6 groups of 8 mice each: a control group and experimental groups, the flaxseed oil group and the fish oil group, for a period of 3 or 6 weeks. At the end of the experiment, 6 and 9 weeks postpartum, the mice were anesthetized with isoflurane and blood and liver samples were collected (under fasting conditions) and stored at −80 °C. Next, their femur and tibia were harvested and treated as follows: the femora were manually cleaned of soft tissue, wrapped in saline-soaked gauze, and stored until analysis at −20 °C (mechanical/Micro-CT testing); the tibiae were fixed immediately for histological analysis.

### 2.2. Diet Preparation and Composition

During the postweaning period, mice were fed a semi-purified diet based on the American Institute of Nutrition (AIN-93) recommendation that was formulated for the growth phase of rodents: 16% fat, 63.5% carbohydrate, and 20.5% protein [16]. The single modification between the diets was the source of fat: (1) fish oil (GRAMSE) rich in n-3 PUFA; (2) flaxseed oil, rich in ALA; and (3) corn oil for the control group (Table 1). All other macro- and micro-nutrients were equal and balanced. The entire diet was homogenized, shaped to form dumplings, and frozen at −20 °C.

### 2.3. Lipid Extraction and Analysis by Gas Chromatography (GC)

Total lipids were extracted using chloroform–methanol solution (2:1, *v*/*v*), and fatty acids; ethyl esters were generated by incubation of lipid fraction with H_2_SO_4_. Heptadecanoin acid was used as an internal standard and was added to the sample prior to its extraction. To analyze the FA content in the oils, diets, serums, and liver samples, we used GC (Agilent Technologies, Santa Clara, CA, USA) equipped with a fused-silica capillary column (60 m × 0.25 mm ID, DB-23, Agilent). The area of each FA peak was recorded using ChemStation software (Agilent Technologies), and FA weights were calculated using the internal standard’s area under the curve and presented as mg/100 mg fat and as percentage (mol.%) of total FAs in each sample [17]. FAs were grouped (sum of mol.% values) into saturated FAs (C8:0, C10:0, C12:0, C14:0, C16:0, C18:0, C20:0, C22:0, C24:0), unsaturated FAs (C16:1n-7, C18:1n-9, C18:1n-7, C18:2n-6, C18:3n-6, C18:3n-3, C20:1n-9, C20:4n-4, C20:5n-3, C22:1n-9, C22:4n-6, C22:6n-3), MUFA (C16:1n-7, C18:1n-9, C18:1n-7, C20:1n-9, C22:1n-9) and PUFA (C18:2n-6, C18:3n-6, C18:3n-3, C20:4n-6, C20:5n-3, C22:4n-6, C22:6n-3). In addition, FAs were grouped into n-3 (C18:3n-3, C20:5n-3 and C22:6n-3) and n-6 (C18:2n-6, C18:3n-6, C20:4n-6 and C22:4n-6) FAs.

### 2.4. Histological Staining of Growth-Plate (GP) Sections

Various staining procedures were used to examine the tibial GPs. Tibiae samples were fixed overnight in 4% paraformaldehyde (PFA, Sigma, St Louis, MO, USA) at 4 °C, followed by 2 weeks of decalcification in 0.5 M EDTA pH 7.4. The samples were then dehydrated, cleared in histoclear (Bar-Naor), and embedded in paraffin blocks. Transverse tissue sections of 5 μm were prepared with Leica microtome (Agentec, Yakum, Israel) for histological staining [15].

The sections were stained in hematoxylin solution for 5 min followed by rinsing in tap water; then the sections were stained with eosin and rinsed again in tap water. For Safranin-O staining, Weigert’s iron hematoxylin solution, fast green solution, and acetic acid were used. The sections were dried and DPX mounting was used for histology.

To stain for alkaline phosphatase, 1-Step NBT/BCIP reagent (Thermo Fisher Scientific, Rehovot, Israel) was used according to the manufacturer’s instructions. Sections were incubated in 0.25% naphthol AS-MX phosphate alkaline solution with fast blue RR salt (Sigma, St Louis, MO, USA), washed with PBS, and incubated with naphthol solution mixture for 1 h at room temperature. The resulting purple, insoluble, granular dye deposit indicated sites of alkaline phosphatase (ALP) activity [18].

### 2.5. Imaging and Measurement of GPs

Stained transverse sections of tibiae were viewed using a light microscopy eclipse E400 Nikon with ×10, ×20, or ×40 objectives, using light filters. Images were captured by a high-resolution camera (Olympus DP 71) controlled by Cell A software (Olympus). Longitudinal median sections from the proximal tibia growth plate were subjected to histological staining and measurements. The thickness of the total GPs, proliferative zone (PZ) and hypertrophic zone (HA) as well as the number of cells were measured using a Cell A software (Olympus) with a measuring tool feature. Measurements were performed on these sections from 5 different animals from each group. In each slide, 10 random locations throughout the GPs were selected and measured [18,19].

### 2.6. Micro-CT

Femur bones were slowly thawed to room temperature before scanning, which was performed using a Skyscan 1174 X-ray computed microtomograph with the following parameters: 50 kV X-ray tube, 800 μA, 0.25 mm aluminum filter at 3000 ms exposure time, and 6.4 μm high spatial resolution. For each specimen, a series of 450 projection images were obtained with a rotation step of 0.4°, averaging two frames, for a total 180° rotation. Flat field correction was performed at the beginning of each scan for a specific zoom and image format. A stack of 2-D X-ray shadow projections was reconstructed to obtain images using NRecon software (Skyscan, Bruker, Belgium) and dynamic image range, postalignment value, beam hardening, and ring-artifact reduction were optimized for each experimental set. Next, to perform a morphometric analysis of the images, we used CTAn software (Skyscan). Detailed 3-D analysis and reconstruction of the sample were performed using the custom software of the micro-CT device, yielding quantitative data. Morphometric parameters were calculated as suggested by the guidelines for bone microstructure assessment. Cortical analysis was performed on a standardized region of interest (ROI) in the mid diaphysis, equidistant from the ends of the bone, containing 150 slices and corresponding to 2.764 mm. The trabecular ROI of the femora consisted of 100 slices, equivalent to 1.86 mm, extending proximally from the end of the distal growth plate (GP) of each bone. Global grayscale threshold levels for the cortical region were between 60 and 255 and for the trabecular region, adaptive grayscale threshold levels between 57 and 255 were used [20].

### 2.7. Mechanical Testing

A three-point bending experiment was conducted in order to characterize the mechanical properties of the bones. Right femora from the mice were tested using a custom-built micro-mechanical testing device. On the day of testing, each bone was slowly thawed and placed within a saline-containing testing chamber that rested on two supports. The supports were located equidistant from the ends of the bone, both in contact with the posterior aspect of the diaphysis. The distance between the stationary supports was set to 8 mm to ensure that the relatively tubular portion of the mid-diaphysis rested on these supports. An initial preload of 0.2 N was applied with a movable prong, which contacted the anterior surface of the bone at a point precisely in the middle between the two supports to hold the bone in place. Force was measured with a dedicated load cell. The experiment was conducted at a constant rate of 400 μm/min up to fracture, as identified by a sudden and significant (>40%) decrease in load.

Force and displacement were recorded at a rate of 10 Hz by a custom designed software program. The resulting force–displacement data of each experiment were used to calculate whole bone stiffness (N/µ), yield load (N) and maximal load (N). The stiffness was calculated as the slope of the linear portion of the load–displacement curve. The yield point was defined as the load at which the load–displacement relationship ceased to be linear [21].

### 2.8. Bone RNA Extraction and RNA-Sequencing

RNA extraction and sequencing were performed on bones from 6-week-old mice. Muscles, tendons, and ligaments were removed with a scalpel. The distal and proximal epiphyses were excised, and the diaphyseal bone marrow was removed by centrifugation at >15,000× *g* for 1 min at room temperature [22]. The resultant hollow bone shafts were individually flash frozen in liquid nitrogen and underwent manual pulverization. Total RNA was extracted and purified from mice ulna and humerus bones using TRI reagent (Sigma, USA) according to the manufacturer’s protocol.

Each RNA sample had a RNA integrity number-RIN > 6, indicating they were of sufficient quality to prepare sequencing libraries, which was performed using INCPM-mRNA-seq, based on the Transeq protocol. Briefly, the polyA fraction (mRNA) was purified from the total RNA, followed by fragmentation and the generation of double-stranded cDNA. Next, end repair, base addition, adapter ligation, and PCR amplification steps were performed. Libraries were evaluated by Qubit (Thermo Fisher Scientific) and TapeStation (Agilent). Sequencing libraries were constructed with barcodes to allow multiplexing of 18 samples in two lanes. Around 20–27 million single-end 60-bp reads were sequenced per sample on an Illumina HiSeq 2500 V4 instrument. Quality control analysis revealed that Q scores of all samples were ~36, Q > 30 is considered a benchmark for quality in NGS.

#### 2.8.1. Bioinformatics

Bioinformatic analyses were performed by the Grand Israel National Center for Personalized Medicine (G-INCPM) research facility, Weizmann Institute of Science, Rehovot, Israel.

Poly-A/T stretches and Illumina adapters were trimmed from the reads using Cutadapt [23]; resulting reads shorter than 30 bp were discarded. Reads were mapped to the M. musculus reference genome GRCm38 using STAR [24], supplied with gene annotations downloaded from Ensembl (with the EndToEnd option and outFilter-MismatchNoverLmax set to 0.04). Expression levels for each gene were quantified with an HTseq-count [25], using the GTF file. Differentially expressed (DE) genes were identified using DESeq2 with the betaPrior, cooksCutoff and independent filtering parameters set to false [26]. Raw *p* values were adjusted for multiple testing using Benjamini and Hochberg’s procedure. Pipeline was run using Snakemake [27].

#### 2.8.2. Gene-Set Enrichment Analysis

Ingenuity Pathways Analysis (IPA) version 4.0 (Ingenuity Systems, Mountain View, CA, USA) was used to search for possible biological processes, canonical pathways, and networks. A detailed description of IPA can be found at www.Ingenuity.com.

To perform significance testing, these steps were followed: (1) the ratio of the number of DE genes from the uploaded data set that was mapped to an IPA pathway was divided by the total number of molecules that existed in the pathway. (2) Fischer’s exact test was used to calculate the probability that the association between the genes in the uploaded data set and the canonical pathway was explained by chance alone. *p* values of < 0.05 were considered significant. (3) The Benjamini–Hochberg procedure was used to calculate the false discovery rate and to correct for multiple testing. Adjusted *p* values of < 0.05 were considered significant. (4) A fold change cutoff of >0.585 and <−0.585 log (differential expression) were applied to all data sets. (5) Z-score analysis was used as a statistical measure of the match between expected relationship direction and observed gene expression of the uploaded dataset. Positive and negative z scores indicated up-regulated and down-regulated pathways, respectively.

### 2.9. Statistical Analysis

All data are expressed as means ± SD. The significance of differences between groups was determined by ANOVA using JMP 12.0.1 Statistical Discovery Software (SAS Institute 2000, Cary, NC, USA). Differences between groups were further evaluated by Tukey–Kramer HSD test, considered significant at *p* < 0.05. Each group that has different letters, differs significantly from the other groups, groups that share letters do not differ from each other.

## 3. Results

To study the effect of dietary n-3 PUFAs from different sources on skeletal development, 3-week-old female C57BL6 mice were divided into three groups and fed isocaloric diets that differed only in their source of fat (Table 1) for 3 or 6 weeks. The diets contained either corn oil (control), flaxseed oil rich in ALA (flaxseed) as a plant source of n-3 PUFA or fish oil rich in EPA and DHA (fish) as an animal source of n-3 PUFA.

### 3.1. Dietary n-3 PUFAs Decrease Fat Accumulation and Alter FA Content in Liver and Serum

GC analysis of FAs showed the differences between the diets with respect to FA composition: the quantity of saturated fatty acids (SFAs) was highest for the fish oil diet and lowest for the flaxseed oil diet. Moreover, the control diet was rich in n-6 PUFAs, while both flaxseed oil and fish oil diets were rich in n-3 PUFAs (Table 2).

PUFAs are highly prone to oxidative degradation [28]. In order to ensure that the FA content of the diets remained as desired throughout the experiments, GC analysis was performed on diet samples that were handled in similar conditions to the actual feeding regime at three different time intervals (time 0, 24 h, 72 h). The results showed no modifications to the FA content of the diets, suggesting that no oxidation had occurred and therefore that the diets were suitable to use (Figure 1).

For fatty acid (FA) profiles, lipids were extracted, concentrated, and analyzed by gas chromatography–mass spectroscopy. Differences in FA content were calculated according to food consumption during the experiment.

The rate limiting enzyme in the LC-PUFA biosynthetic pathway is delta-6 desaturase, which is mostly expressed in the liver both in humans and in mice [29]. Thus, the effects of the different FAs on metabolism and growth and bone parameters also depend on the modifications that occur in the liver.

Hepatic fat levels as measured by GC analysis were 1.6 times lower in the mice that were fed the flaxseed oil diet and two times lower in mice that were fed the fish oil diet compared to the control mice on the corn oil diet (Figure 2A), suggesting that consumption of dietary n-3 PUFAs reduced fat accumulation in the liver. Furthermore, dietary n-3 PUFA enrichment increased the hepatic levels of n-3 PUFAs; mice in the flaxseed oil group had 1.7 and 7.11 times higher levels of n-3 PUFAs in the liver compared with mice in the fish oil group and control mice, respectively (Figure 2B). The total n-6 PUFAs levels in the liver were significantly lower in the fish and flaxseed groups compared with the control group: 3.8- and 6.3-fold, respectively (Figure 2C). As a result of these effects, the n-6 to n-3 ratio in the liver of mice in the flaxseed and fish groups was dramatically reduced (Figure 2D).

We also assessed the FA content in the serum in order to investigate its correspondence to the FA content in the liver. Total serum lipid content was significantly lower in the fish and flaxseed groups as compared with the control (Table 3), resembling the results obtained for the liver. This indicates that the serum lipid levels that the cells were exposed to were significantly affected by the experimental diets. Both treatment groups, flaxseed and fish, displayed significantly lower n-6:n-3 FA ratios compared with the control (11.5 and 15.5 times lower, respectively).

Similar to the result in the liver, consumption of both diets containing n-3 PUFAs was comparably associated with a significant reduction in serum AA levels as compared with the control group. However, the fish group had significantly higher serum AA levels, as compared with those in the flaxseed group (Table 3).

Comparative analyses between the hepatic FA profiles of the mice that were fed one of the three diets and the FA profiles of the diet (Table 4) demonstrated the modifications occurred in the liver. For instance, in the control group that consumed a corn-oil-based diet rich in LA (97.45% of the PUFAs in the diet) and low in ALA (1.7% of the PUFAs in the diet), hepatic levels of n-6 PUFAs were 32.67% higher compared with hepatic n-3 PUFAs (comparable to the diet). However, higher levels of DHA were detected in the liver of the control group compared to the diet, probably due to the conversions, albeit inefficient, of ALA to its derivatives.

The estimated activity of delta-5 desaturase and delta-6 desaturase was significantly reduced in the flaxseed oil group as compared to the control group, suggesting that the conversion of LA to AA was inhibited by the presence of ALA (Figure 2E,F). This is supported by the apparent reduction of AA from 18.89% in the control to 4.63% in the flaxseed oil and to 11.4% in the fish oil diet group. Interestingly, although the percentage of LA was 2.6 times higher in the flaxseed oil diet compared with the fish oil diet, in which the more mature form of n-3 PUFAs was abundant, mice that were fed the flaxseed oil diet had a lower percentage of hepatic AA, n-6 derivative (2.4 times lower). Furthermore, while the percentages of EPA and DHA (n-3 derivatives) in the liver of mice that were fed the fish oil diet corresponded to their percentages in the diet, in mice that were fed the flaxseed oil diet, hepatic levels of EPA and DHA were high despite their low levels in the diet. These results suggest that ALA enrichment of the diet (as in the flaxseed diet) affects the biosynthetic pathway of PUFAs by diverting it toward n-3 PUFA conversions from ALA to EPA and DHA rather than to n-6 PUFA conversions from LA to AA.

### 3.2. Effect of Omega-3 from Different Sources on Growth Pattern and Food Consumption

Measurements of food intake, body weight, and tail length showed a standard growth pattern with no significant differences between the three groups (Figure 3A–D). The growth pattern corresponded to the standard, with a rapid growth rate from 3 to 6 weeks of age and a slower rate from 6 to 9 weeks of age upon reaching sexual maturity (Figure 3C,D). Femur length, an additional parameter of skeleton longitudinal growth, did not differ between the groups, either at 6 weeks old, or at 9 weeks old (Figure 3E). Taken together, these results show that the different fat sources in the diet did not affect food consumption and the general growth pattern in the young mice.

Since longitudinal bone growth originates from the growth plate (GP) [30], we further evaluated possible microstructure differences between the groups. Safranin-O staining analyses revealed that the GPs of mice in the flaxseed oil and fish oil groups had a regular structure of all the zones (resting, proliferating, and hypertrophic) with normal cell morphology (Figure 4A). A key component of the hypertrophic matrix is ALP, an enzyme that is implicated in the mineralization process [31]. ALP staining based on colorimetric insoluble substrate of the enzyme, thus indicating ALP activity in-situ. ALP activity did not differ between the groups (Figure 4).

At the age of 9 weeks, the total GP thickness as well as the thickness of the specific regions was significantly higher in both flaxseed and fish groups compared with the control mice (Figure 4B). At the age of 6 weeks, the total GP thickness of mice that were fed the flaxseed oil or fish oil diet was 11% and 6% higher compared with mice in the control group, respectively (Figure 4A). These differences were reflected in the PZs of the mice, while no differences were indicated between the groups in the HZ. No significant differences were found between the groups either at 6 weeks or at 9 weeks of age in the number of cells in the total GPs, PZs, and HZs (Table 5). These results imply that diet rich in ALA or in EPA and DHA does not affect chondrocytes proliferation, but enhances matrix production, which leads to a thicker GP.

Some support for these results could be found in an in-vitro study that showed that treating chondrocytes with n-3 PUFAs resulted in an increase in cells differentiation and matrix production [15]. Furthermore, differences in GP width were not reflected in the femoral length, probably due to different measurement scales; while femoral length was measured on a scale of mm, GP width was measured using microscopic software on a scale of µm that allows detection of slighter changes.

### 3.3. The Effect of Diets Rich in n-3 PUFAs on Bone Quality

#### 3.3.1. Diets Rich in n-3 PUFAs Improve Bone Mechanical Properties

The skeleton plays a critical role in bearing functional loads, which depends on the bones’ capacity to withstand fractures; it is therefore important to assess their mechanical properties. The biomechanical properties of the femur were evaluated in a three-point bending experiment (Table 6). At 6 weeks of age we found that most of the tested parameters were significantly higher in mice that were fed the fish oil diet. A similar trend was observed in mice that were fed the flaxseed oil diet, yet to a lesser degree. Specifically, maximal load was substantially higher in both the flaxseed and fish oil groups compared with the control group. Additionally, femora of mice fed a fish oil diet were stiffer that those of the control group (Table 6). These parameters measuring the stiffness, plastic, and elastic properties of the bone indicate the ability to bear the load. No differences were found between the groups at 9 weeks of age.

#### 3.3.2. Diet Containing Plant Source of n-3 PUFAs (Flaxseed Oil) Improved Trabecular Bone’s Micro-Architecture

The comparison between the groups demonstrated significant differences in trabecular bone micro-architecture. As expected, the trabecular number (Tb.N) for all the groups decreased slightly when the mice reached 6 to 9 weeks of age as a result of bone maturation [32] (Table 6). Both the 6- and 9-week-old mice in the flaxseed oil group had a significantly larger percent of bone volume (BV/TV), which increased by 30%, and a higher Tb.N, which increased by 20% compared with the control group (Table 6, Figure 5). The group that was fed the fish oil diet had a slightly higher BV/TV and Tb.N compared with the control group, but this was not significant. Trabecular thickness and separation did not differ between the groups (Table 6). These data suggest that the presence of n-3 PUFAs from a plant source contributes to the process of bone development and optimizes characteristics of trabecular bone.

#### 3.3.3. Diet Containing n-3 PUFAs from an Animal Source (Fish Oil) Increased BMD

We found that diets rich in n-3 PUFAs had no significant effect on the geometrical features of the cortical bone (Table 6), although a trend towards increased cortical area fraction was detected in the mice consuming the flaxseed-based diet. Results showed that the BMD of 9-week-old mice on the fish oil diet was significantly higher compared with the mice on the flaxseed oil and control groups. At the age of 6 weeks, these differences were still not significant (Table 6), suggesting that the diet containing an animal source of n-3 PUFAs (fish oil) resulted in a higher accumulation of mineral in the young developing bones.

### 3.4. The Effect of Different Sources of n-3 PUFAs on Bone’s Transcriptional Regulation

To verify the underlying mechanisms of the different n-3 PUFA effects on bone phenotype of 6-week-old mice, we applied the pairwise analysis of the signals from 19,040 genes. We identified 763 genes that were differentially expressed (DE) in at least one of the following comparisons: flaxseed versus control, fish versus control, or flaxseed versus fish with a *p* value below 0.05 and a log-2-fold change greater than 0.585 (Appendix A).

Figure 6A summarizes the number of DE genes for each comparison. Twenty-two genes were up-regulated and 33 were down-regulated in the flaxseed group compared with the control group, 619 were up-regulated and 77 down-regulated in the fish group compared with the control group, and 34 were up-regulated and 227 were down-regulated in the flaxseed group compared with the fish group (Figure 6B).

#### IPA Pathway Enrichment Analysis of Genes Related to Dietary n-3 PUFAs

The top signaling pathway related to bone development was clarified (Table 7) in the transcriptome analysis. The flaxseed-oil-based diet was enriched for genes implicated in the circadian rhythm, with about 20% of the pathway genes either up- or down-regulated compared with the control diet. The fish-oil-based diet altered the cyclin and cell-cycle pathways, with 40% of the pathway genes either up- or down-regulated compared with the control diet; and the comparison between the flaxseed oil diet and the fish oil diet indicated changes associated with the heme biosynthesis pathway.

To further investigate the functional influence of the fish oil diet on cell-cycle progression, we used the IPA functional analysis tool, which indicated an activation during interphase and inhibition of re-entry to the cell cycle. On the basis of these two analyses, we inferred that the fish oil diet inhibited cell-cycle progression. Figure 6C presents the up- and down-regulated DE genes that were classified as involved in pathways related to cell cycle. Despite the observed up-regulation of cyclin-E, B, A which promote cell-cycle progression, the G1/S check point pathway was significantly down regulated, this is due to the complexity of the cell cycle process that consists of many regulators that influence its direction of action. Several genes that are known to induce cell-cycle arrest were up-regulated in the fish oil group, including RB1, p18, and p107 as well as the transcriptional repressors E2F-4,7,8. Moreover, we observed that the expression of Cyclin Dependent Kinase Inhibitor 1A (CDKNA1, also known as p21), a negative regulator of cell-cycle progression, was 1.5 and 1.7 times higher in the fish oil group and flaxseed oil group compared with the control group, respectively. However, it is evident that the fish oil diet influenced cell-cycle genes to a higher extent compared with the flaxseed oil and control diets.

We also used the IPA functional analysis tool to study the canonical pathways related to bone development; these included: the role of osteoblasts, osteoclasts, and chondrocytes in rheumatoid arthritis, sonic hedgehog signaling, role of macrophages, fibroblasts and endothelial cells in rheumatoid arthritis, osteoarthritis pathway and Wnt β-catenin signaling, which were significantly affected by one or more of the interventions (Figure 6D).

Results show that while the matrix protein collagen type 5α3 (COL5A3) expression was up-regulated in the flaxseed oil group compared with the control group, the expression of collagen type 27α1 (COL27A1) and 24α1 (COL24A1) was down-regulated in the fish group compared with the control group. The comparison of Matrix Gla protein (Mgp) expression between the flaxseed oil and fish oil groups showed an up-regulation in the former. The matrix metalloproteinases ADAMTS-4, 5, and 15 were up-regulated in both the flaxseed oil and fish oil groups compared with the control group. Osteoclastogenic markers, calcitonin receptor (Calcr), and the nuclear factor of activated T-cells, cytoplasmic 2 (Nfatc2), were significantly down-regulated in the fish group and flaxseed group compared with the control group.

Periosteal osteoblast precursors are potential sources of new osteoblasts. Differentiation of mesenchymal stem cells toward tissue-specific lineages to either osteoblasts or adipocytes is sensitive to environmental conditions such as dietary changes. Results showed that consumption of fish oil diet led to an up-regulation in PPARγ gene expression compared with the control group (Figure 6D).

Comparative enrichment analysis of significantly regulated canonical pathways indicated an overlap between the different groups. Both diets containing oils rich in n-3 PUFAs resulted in an alteration to the circadian rhythm pathway (Figure 6E). To better understand the involvement of n-3 PUFA and circadian rhythm in bone development, we focused on the specific genes affected and indicated consistent patterns of gene expression. The expression of ARNT-like protein-1 (Bmal1) was significantly up-regulated, and the expression of the Nuclear Receptor Subfamily 1 Group D Member 1 (Nr1d1), Period2 (Per2), Period3 (Per3), Cryptochrome2 (Cry2), D-site albumin promoter binding protein (Dbp), and basic-helix-loop-helix (Bhlhe41) was down-regulated.

## 4. Discussion

This study describes the role of different dietary sources of n-3 PUFAs in skeletal development and bone quality. We found that dietary n-3 PUFAs contributed to improved mechanic and morphometric properties of the bone, hence to improved bone quality. Notably, the positive effects depend on the sources of n-3 PUFAs. Evidence presented over recent has shown that LC-PUFAs, especially n-3 PUFAs such as EPA and DHA, are beneficial for bone health. Findings from observational and randomized controlled trials indicated that LC-PUFAs can enhance bone formation in adult and older men and women, increase PBM in adolescents, and reduce bone loss of postmenopausal and osteoporotic women as measured by dual energy x-ray absorptiometry (DEXA) [3,33,34]. Factors influencing initial bone development are of great importance because of the implications for bone health later in life. Women in particular are susceptible to bone mineral loss and compromised bone structure at a greater rate as they approach menopause and loss of endogenous estrogen production. We therefore selected female mice for our study and propose our findings as the basis for a preventative approach to promoting bone quality. Androgens and estrogens are considered as major regulators of gender differences in bone metabolism. Regarding the consumption of PUFAs, Lau et al., suggested that the dietary recommendations for PUFAs intake need to be gender-specific [35].

### 4.1. Dietary n-3 PUFAs Decrease Hepatic and Serum Fat Levels

The different dietary sources of n-3 PUFAs contain FAs that differ in their biochemistry and their metabolic pathways. The FA content in the diet can modulate the composition of stored and structural lipids, including the FA profiles of tissues [36]. Our study demonstrates that dietary n-3 PUFAs at a physiological concentration (16%) with no further challenging decrease fat levels in the liver and serum. Mice that were fed flaxseed oil or fish oil diets had significantly lower levels of total hepatic fat. Studies that have investigated the effect of n-3 PUFAs on hepatic FA content usually focused on alcoholic liver steatosis. For instance, Huang et al. reported that the fat-1 mice exhibited significantly reduced hepatic fat levels. They suggested that endogenous n-3 PUFAs have protective effects against alcoholic liver disease [37].

We also show the effect of n-3 PUFAs on hepatic FA content. The hepatic levels of n-3 PUFA in the mice that were fed the flaxseed oil diet were higher than those whose n-3 PUFA intake came from fish and from those in the control group. The liver is a major site of LC-PUFAs biosynthesis [38], hence, their deposition is dependent not only on the intake of EPA and DHA but also on the intake of their precursor, ALA. The flaxseed oil diet, similar to the control diet, did not contain n-3 LC-PUFAs; nonetheless, the mice that were fed the flaxseed oil diet had higher levels of EPA as well as DHA in the liver compared with the control mice, demonstrating that ALA enrichment resulted in improved efficiency of ALA conversion to its derivatives. Still, the levels of hepatic EPA and DHA in mice from the flaxseed oil group were lower compared with mice in the fish oil group, possibly due to preferential use of ALA as an energy source, resulting from beta-oxidation or because of the inefficient conversion of ALA to EPA and DHA. Interestingly, despite comparable levels of dietary LA (18:2 n-6), the mice that were fed the flaxseed oil diet had the lowest level of the n-6 LC-PUFA, AA (20:4 n-6) in the liver compared with mice in the other groups. The differential metabolite composition may result from the competition between LA and ALA for the rate limiting enzyme delta-6 desaturase for their conversion to n-6 or n-3 LC-PUFAs, respectively. In the competition for the enzyme, a preference has been shown toward ALA [39], suggesting that the high amount of ALA in the flaxseed oil diet reduced production of the n-6 LC-PUFAs.

Among other polyunsaturated free fatty acids (FFA), n-3 PUFAs have been recognized for their beneficial effects on the hepatic lipid metabolism as well as on the serum lipid profile as signaling molecules [40,41]. In this study, plasma was collected under fasting conditions, therefore we assume that the majority of FAs were present in serum in their free form, as FFAs. We focused on FFAs since FFA receptor 4 (FFAR4) is expressed in bone cells and preferentially binds n-3 PUFAs [15,42]. In our study it was expressed in the bone samples of all groups without significant differences. FFAR4 was shown to stimulate bone formation and suppress bone resorption upon activation by n-3 PUFA [43]. The activation of the FFAR4 pathway is possibly the underlying connection between nutrition, lipid metabolism, and bone metabolism and might be crucial to the protective effects of dietary n-3 PUFAs on bones.

### 4.2. Diet Rich in n-3 PUFAs Affects Postnatal Skeletal Development and Bone Characteristics

Micro-CT analysis suggests that dietary n-3 PUFAs affect bone quality by influencing different osseous tissues; while flaxseed oil improved trabecular bone properties, fish oil enhanced mineralization. This finding is supported by Lukas et al.’s recent work, which showed that rats that were fed a high-fat diet (26% Kcal), rich in DHA, had higher tibial BMD. In addition, rats on a high-fat diet (26% Kcal), rich in ALA, had an improved trabecular bone micro-architecture [7]. Our study exhibits analogous results despite using a balanced diet (16% Kcal) in a different animal model.

The observed differences in bone micro-architecture and mineralization are expected to affect the mechanical behavior of long bones [44]. Lau et al. suggested that a reduction of the n-6:n-3 PUFA ratio and an increase in EPA and DHA are linked to greater bone strength in healthy young fat-1 male and female mice [33,35]. We found that diets rich in n-3 PUFAs improve bone mechanical properties in 6-week-old mice. However, this result was not indicated in 9-week-old mice, probably due to limitations in the three-point bending test of the whole bone. Reliable measurement of material properties requires the use of samples with well-defined geometry, like rectangular beams. Such samples are difficult to produce from thin cortices like those of mice. When whole bones are tested, results are dependent upon both the geometry of the bone and the mechanical properties of the material [21]. Osteoblasts and adipocytes derive from a shared pool of bone marrow mesenchymal stem cells, while metabolic microenvironment, such as nutrition, regulates bone marrow progenitor cells differentiation. Several studies showed a tradeoff between bone and fat mass, with the greater differentiation of adipocytes at the expense of osteoblasts thereby leading to reduced bone mass. In this study we evaluated marrow fat accumulation using tibial histological sections. Thus, we conducted a double blind evaluation based on visualization by eight independent examiners on six different slides from each group. Results, when calculated and quantified, show no significant differences between the treatments. Despite the impression of higher level of fat globules in the control group samples as compared to the n-3-enriched groups, we could not establish this fact. We assume that upon higher levels of fat in the diets these differences could become more pronounced.

### 4.3. Transcriptional Regulation of Bone by Different Sources n-3 PUFAs

To better understand the impact of n-3 PUFAs on the transcriptome of the developing bone, we performed RNA-seq analysis of the bones of mice whose PUFA intake originated from different dietary sources. Although a few studies have used RNA-seq analysis to reveal the DE genes in skeletal tissue [45,46,47], as far as we know, this is the first study that uses RNA-seq to study the effect of dietary n-3 PUFAs on bone transcriptome in vivo.

Several of the findings in the transcriptome analysis were as expected: the most highly expressed genes (>100,000 counts per sample) were collagen-encoding genes, which are the major bone matrix protein expressed by osteoblasts (COL1a, COL1b) [48]. Osteopontin (Spp1), Cathepsin K (Ctsk), Osteocalcin (Bglap), TRAP (Apc5), bone sialoprotein (Ibsp), and alkaline phosphatase (Alpl) are matrix proteins produced by osteoblasts and osteoclasts and were also highly expressed (>1000 counts per sample) [49]. Osteocytes typical genes were also indicated in this analysis, including ORP150, CapG, DMP1 and PDPN.

N-3 and n-6 PUFAs are ligands of PPARγ, which is known to influence distinct target genes in various cell types, including bone cells [50]. Furthermore, eicosanoids and lipid mediators produced from AA, EPA, and DHA also bind and regulate PPARγ [12]. One of the actions of PPARγ is to physically inhibit the translocation of NFκB to the nucleus. This might be the mechanism by which n-3 PUFAs perform their anti-inflammatory activity [51]. Additionally, PPARγ was found to stimulate adipocyte differentiation at the expense of osteoblast differentiation in bone marrow mesenchymal stem cells [51]. Although it can be argued that PPARγ activation increases bone resorption, its up-regulation in the mice that were fed the fish oil diet was not accompanied by an increase in bone resorption genes or decreased BMD. In mice on the flaxseed oil diet, nuclear factor of activated T-cells 2 (Nfatc2), a bone resorption related gene was down-regulated compared with the mice on the fish oil diet and the control mice. The calcitonin receptor (Calcr) is expressed by mature osteoclasts and considered a bone resorption gene [52]; therefore its down-regulation in the fish oil group compared with the control group might explain the higher BMD levels. Moreover, genes associated with bone formation were not differentially expressed.

Interestingly, it has been suggested that the inhibition of PPARγ prevents the action of n-3 PUFAs during cell differentiation [53]. This hypothesis led us to focus on the role of n-3 PUFAs in cell-cycle progression and its relationship to the observed skeletal phenotype (Figure 7).

### 4.4. Dietary n-3 PUFAs Alter Circadian Clock Gene Expression in Bone

We observed a remarkable trend in both of the groups that were fed n-3 PUFA diets: the core circadian clock genes were altered in both. The presence of a peripheral clock in bone cells has increasingly been recognized as a major regulator controlling cellular functions [54]. Whereas the circadian clock in the suprachiasmatic nucleus (SCN) in the brain is mainly responsive to light, peripheral clocks are influenced by various factors, among them dietary macronutrients [55]. Current understanding of the circadian clock’s role in bone metabolism mainly draws from in vitro studies as well as from transgenic models. Here, we show for the first time, the relationship between dietary fats and modifications in the expression of circadian clock genes in the bone.

Genes related to the circadian rhythm participate in the regulation of bone resorption directly through osteoclastogenesis inhibition and indirectly through influencing the osteoblasts’ regulatory activity of osteoclastogenesis [54,56,57]. The classical clock machinery comprises two positive and two negative elements: Bmal1 and Clock are the former; Per and Cry are the latter [58]. These genes were found to be involved in the differentiation and proliferation of osteoblasts and osteoclasts. We found that mice that were fed a diet containing either flaxseed oil or fish oil had a greater expression of Bmal1 in the long bones and a lower expression of the clock genes Bhlhe40, Cry2, Per2, Per3, and Dbp, as compared with the control mice.

These results corroborate previous studies showing improved bone morphology along with changes in the expression of circadian clock genes. Takarada et al. reported that osteoblast-specific knockout (KO) of Bmal1 in mice resulted in significantly lower BV/TV, trabecular thickness, and Tb.N. Moreover, in the femur of Bmal1 KO mice, an increase was predominantly found in bone resorption marker genes but not in the expression of bone-formation related genes [57]. Our results match the findings described by Takarada: up-regulation of Bmal1 was accompanied by improved trabecular bone properties and lower expression of genes associated with bone resorption such as calcitonin receptor and Nfatc2; and no significant differences in bone-formation genes. These data further support the hypothesis that Bmal1 up-regulation in osteoblasts might inhibit bone resorption. The effects of Per and Cry core circadian genes on osteoblasts are the opposite of those that Bmal1 and Clock exert. Osteoblast-targeted deletion of Per genes (Per1 and Per2) or Cry genes (Cry1 and Cry2) was associated with increased bone mass, higher bone-formation rate, and lower osteoclast activity than that observed in control mice [59]. The morphological effects attributed to these genetic manipulations correlate with our data on bone phenotypes and down-regulation of Per and Cry genes (Figure 7).

In a Bmal1 KO mice model, Takarada et al., indicated hypersensitivity to 1,25(OH)2D3 and a common thread among the clock system and 1,25(OH)2D3 signal in osteoblasts [57]. Moreover, in an in vitro experiment they found that Bmal1-deficient osteoblasts support osteoclastogenesis to a higher extent, suggesting that the bone modeling–remodeling process is regulated by an osteoblastic clock system through a mechanism related to the modulation of 1,25(OH)2D3-induced Rankl expression in osteoblasts. Additionally, the VITamin D and OmegA-3 TriaL (VITAL), is an ongoing clinical research in over 25,000 men and women across the United States. The main goal of VITAL is to determine whether vitamin D and/or n-3 PUFAs can prevent cancer, heart disease, and stroke. However, other health outcomes, such as risk for bone fractures, are now being examined as they might have potential therapeutic effect on bones [60].

Proliferation and differentiation of osteoblasts and osteoclasts and the cellular interaction between them are vital to the regulation of bone remodeling. Both VDR and PPARγ are ligand-activated nuclear transcription factors that are instrumental to bone health. In addition to its function in lipid metabolism, PPARγ is directly linked to the peripheral circadian clock [61]. PPARγ transcription is driven by Bmal1, and PPARγ in turn activates the transcription of Bmal1. In accordance with our RNA-seq results indicating activation of cell differentiation by n-3 PUFAs, PPARγ and 1,25(OH)2D3 were also reported to have antiproliferative effects. Therefore, it can be assumed that dietary n-3 PUFAs can influence the circadian clock through PPARγ and VDR, which might eventually lead to alterations in skeletal development.

Our findings show a link between the consumption of n-3 PUFAs and bone circadian rhythm and improved bone phenotype. Together with evidence from the bone-specific KO of clock genes, we suggest that the pivotal role of dietary n-3 PUFAs in regulating bone quality is mediated by the peripheral circadian clock system.

The circadian clock pathway involves the negative autoregulation of the Per and Cry genes via the inhibition of the activators Bmal1 and Clock through the Per and Cry proteins [62]. An additional negative feedback that represses Bmal1 expression is mediated by the Nr1d1 protein, which is itself induced by Clock/Bmal [55]. One of the actions of circadian repressors Cry1 and Cry2 is modifying transcriptional activity through interacting with nuclear receptors such as PPARγ and vitamin D receptor, both are ligand-activated nuclear transcription factors that are crucial for bone health [63]. N-6 and n-3 PUFAs, eicosanoids, and other lipid mediators produced from AA, EPA, and DHA can bind and regulate PPARγ, which can disturb the translocation of NFκB to the nucleus. This action might be a mechanism by which n-3 PUFAs perform their anti-inflammatory activity [51].

Several molecular links between circadian clock and cell-cycle genes are known, one of which is shown; Nr1d1 binds to the same element present in p21 promotor and inhibits p21 transcription, leading to cell-cycle progression [64,65]. The mechanism by which n-3 PUFAs might regulate bone formation is through down-regulation of Nr1d1, which results in increased levels of p21, inhibition of cell cycle, and increased differentiation. Negative cell-cycle regulators (such as pRB and p107) inhibit the transcription of cell-cycle genes and promote differentiation [66,67]. Moreover, p107 is a vital component mediating the antiproliferative activity of 1,25(OH)D3; hence its up-regulation is thought to promote cell differentiation [68].

## 5. Conclusions

Our results emphasize the importance of integrating various sources of n-3 PUFAs into the diet from an early age. Our findings also suggest that Omega-3 may be an important nutrient to include in diets that promote bone health especially in view of the general population’s tendency over the past three decades toward excessive consumption of n-6 PUFAs [69].

## Figures and Tables

**Figure 1 nutrients-12-03494-f001:**
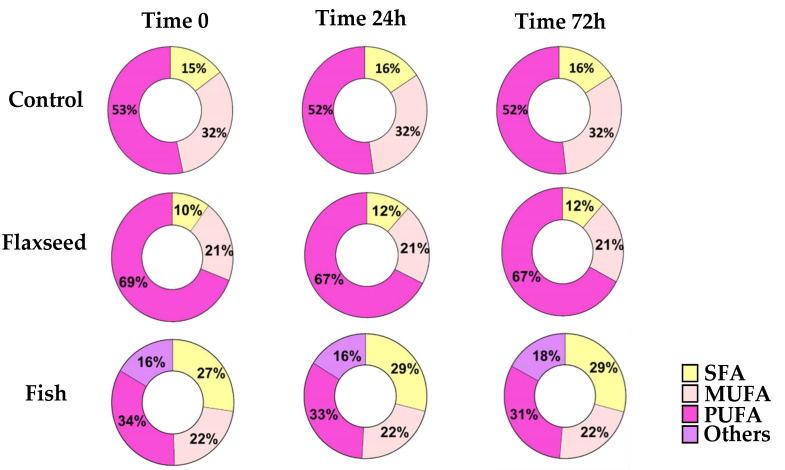
For fatty acid (FA) profiles, lipids were extracted, concentrated, and analyzed by gas chromatography–mass spectroscopy. FA profiles of the diets at three different time intervals (time 0, 24 h, 72 h) for SFAs (saturated fatty acids); MUFAs (mono-unsaturated fatty acids); and PUFAs (polyunsaturated fatty acid).

**Figure 2 nutrients-12-03494-f002:**
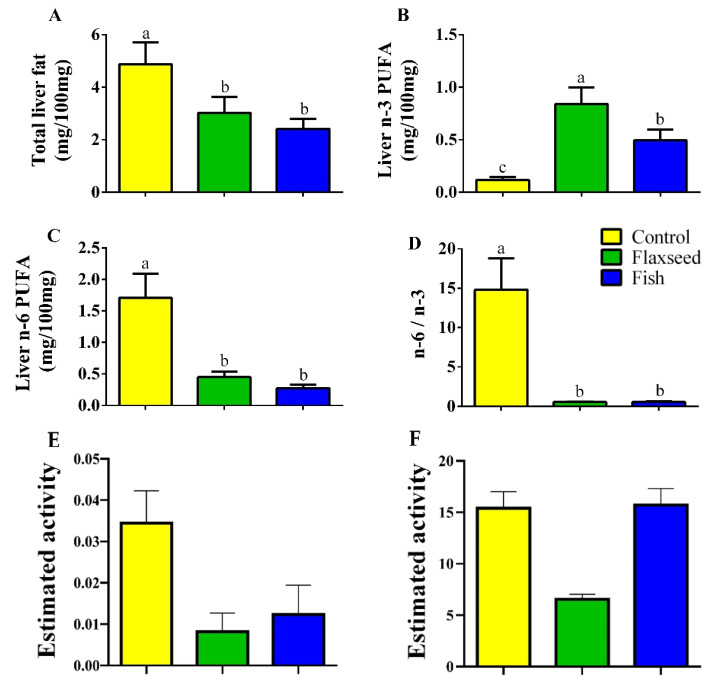
Hepatic FAs of 9-week-old mice. Lipids were extracted from liver samples and FA composition was analyzed by gas chromatography (GC). (**A**) Total hepatic fat levels (mg/100 mg liver). (**B**) Total hepatic n-3 PUFAs. (**C**) Total hepatic n-6 PUFAs. (**D**) Hepatic n-6 to n-3 ratio. (**E**) Desaturase activity was estimated as the product/precursor ratio of individual FA according to the following: delta-6 desaturase (18:3 n-6/18:2 n-6) and (**F**) delta-5 desaturase (20:4 n-6/20:3 n-6). Values are expressed as means ± (SD) of *n* = 8 mice/group; different superscript letters indicate significant difference (*p* < 0.05) which was determined using a one-way ANOVA followed by Tukey’s test.

**Figure 3 nutrients-12-03494-f003:**
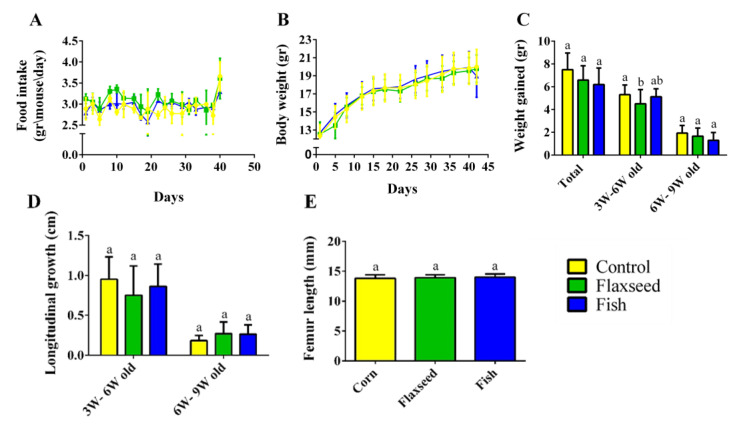
Food consumption and growth patterns. (**A**) Food intake (gr/mouse/day). (**B**) Weight during the entire experiment (gr). (**C**) Growth pattern represented by the total weight gain during two periods: 3–6 weeks of age and 6–9 weeks of age. (**D**) Longitudinal growth of the tail (cm). (**E**) Femur length measured by micro-CT (mm). Values are expressed as means ± SD of *n* = 8 mice/group; different superscript letters indicate significant difference (*p* < 0.05) which was determined using a one-way ANOVA followed by Tukey’s test.

**Figure 4 nutrients-12-03494-f004:**
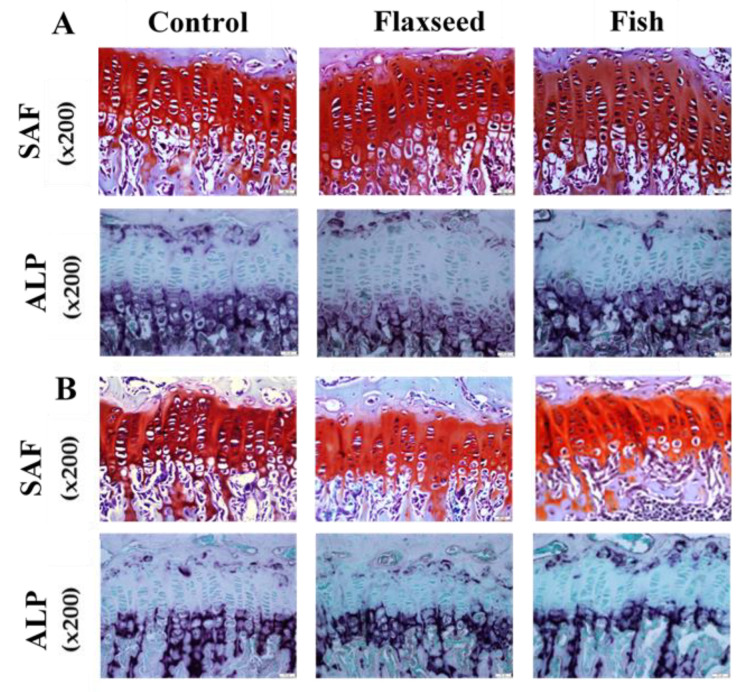
Histological evaluation of the tibial growth plates (GPs) of 6- and 9-week-old mice was performed for the analysis of GP morphology. Transverse tissue sections of 5 μm were prepared by microtome. (**A**) SAF-Safranin-O, and ALP-Alkaline phosphatase staining of representative tibial GPs of 6-week-old mice and (**B**) 9-week-old mice for the three different groups.

**Figure 5 nutrients-12-03494-f005:**
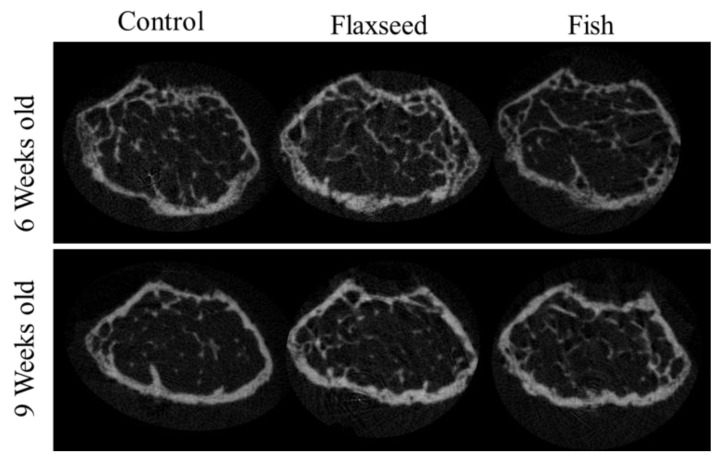
3D images of trabecular bone from 6- and 9-week-old mice.

**Figure 6 nutrients-12-03494-f006:**
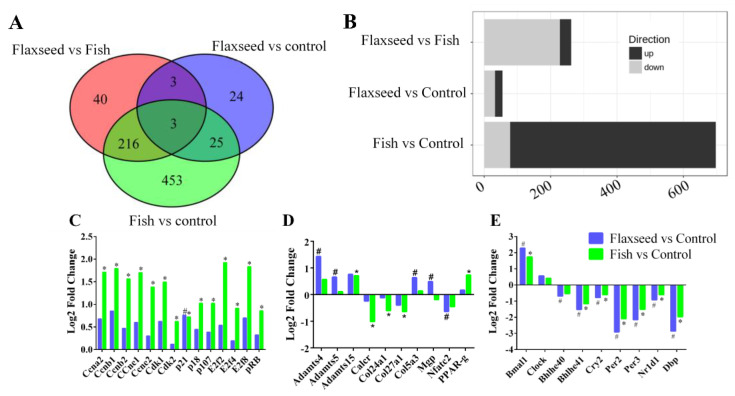
(**A**) Venn diagrams indicating the intersection of differentially expressed (DE) genes that have 1.5 or greater expression (*p* < 0.05) in bone for each comparison. (**B**) Bar plots summarize the number of DE genes for each comparison. (**C**) DE genes related to the cell-cycle canonical pathway, (**D**) to bone modeling–remodeling processes, and (**E**) to the circadian rhythm pathway. Differential expressed cutoff was set at *p*-value ≤ 0.05, log-2-fold change ≥0.585, min counts ≥30. Raw *p-*values were adjusted for multiple testing using Benjamini and Hochberg’s procedure. Sign of * and ^#^ indicate statistically significant differences. Data were analyzed using Ingenuity Pathways Analysis (IPA) (QIAGEN Inc., Petach-Tikva, Israel).

**Figure 7 nutrients-12-03494-f007:**
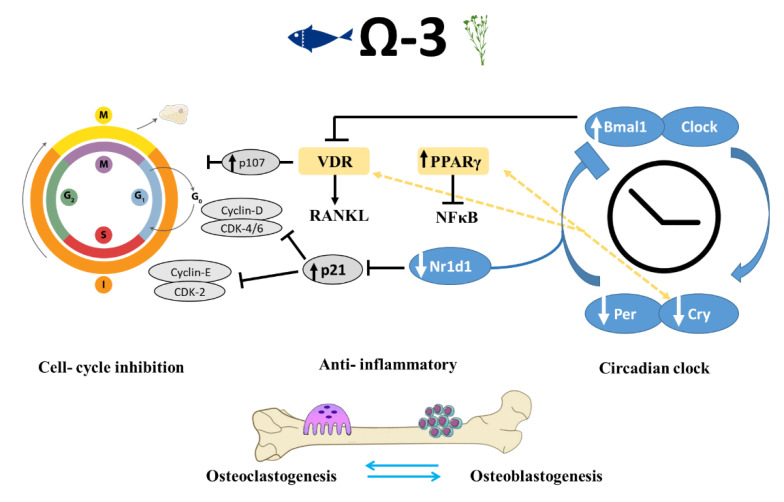
Schematic view of mechanisms implicated in the regulation of bone-formation process by circadian clock and cell-cycle genes in mice that were fed n-3 PUFAs. Shown are the pathways identified in this and previous studies, suggesting that n-3 PUFAs are linked to bone remodeling at multiple levels via peripheral circadian clock and cell-cycle genes, as well as through their anti-inflammatory activity. VDR-vitamin D receptor, RANKL-Receptor activator of nuclear factor kappa-Β ligand, PPAR-peroxisome proliferator-activated receptor, NFKB- nuclear factor kappa-Β.

**Table 1 nutrients-12-03494-t001:** Macronutrient and mineral composition of the three experimental diets.

Ingredient	g/kg diet	Kcal/kg
Cornstarch	397.486	1590
Casein (≥85% protein)	200	800
Dextrinized cornstarch (90–94% tetrasaccharides)	132	528
Sucrose	100	400
Corn oil\flaxseed oil\fish oil	70	630
Fiber	50	
Mineral mix (AIN-93G-MX)	35	
Vitamin mix (AIN-93-VX)	10	
L-Cystine	3	12
Choline bitartrate (41.1% choline)	2.5	
Tert- butylhydroquinone	0.014	

**Table 2 nutrients-12-03494-t002:** Differences in FA content between the diets.

Fatty Acids (gr\mouse)	Diet		
	Control	Flaxseed	Fish
SFAs	1.68	1.05	3.49
MUFAs	3.21	1.88	2.65
PUFAs	5.87	7.69	3.54
Total n-6 PUFAs	5.74	1.77	0.5
Total n-3 PUFAs	0.12	5.89	3.01

**Table 3 nutrients-12-03494-t003:** Hepatic and serum FAs profiles.

Fatty Acid	Liver			Serum		
	Control (mg/100 mg)	Flaxseed (mg/100 mg)	Fish (mg/100 mg)	Control (mg/mL)	Flaxseed (mg/mL)	Fish (mg/mL)
**Total n-3**	**0.118 (0.02) ^b^**	**0.839 (0.15) ^a^**	**0.494 (0.10) ^b^**	**0.100 (0.03) ^b^**	**0.535 (0.07) ^a^**	**0.522 (0.09) ^a^**
18:3 ALA (n-3)	0.011 (0) ^b^	0.539 (0.12) ^a^	0.015 (0) ^b^	0.005 (0.00) ^b^	0.232 (0.03) ^a^	0.014 (0.00) ^b^
20:5 EPA (n-3)	0.002 (0) ^b^	0.158 (0.02) ^a^	0.157 (0.04) ^a^	0.003 (0.00) ^c^	0.184 (0.03) ^b^	0.241 (0.03) ^a^
22:6 DHA (n-3)	0.082 (0.02) ^c^	0.132 (0.02) ^b^	0.316 (0.05) ^a^	0.066 (0.03) ^b^	0.102 (0.01) ^b^	0.252 (0.05) ^a^
**Total n-6**	**1.710 (0.38) ^a^**	**0.449 (0.08) ^b^**	**0.274 (0.05) ^b^**	**1.381 (0.36) ^a^**	**0.587 (0.08) ^b^**	**0.430 (0.09) ^b^**
18:2 LA (n-6)	1.282 (0.34) ^a^	0.376 (0.07) ^b^	0.176 (0.04) ^b^	0.820 (0.31) ^a^	0.441 (0.07) ^b^	0.240 (0.06) ^b^
20:4 AA (n-6)	0.343 (0.05) ^a^	0.059 (0.01) ^b^	0.086 (0) ^b^	0.480 (0.04) ^a^	0.101 (0.01) ^c^	0.147 (0.03) ^b^
**SFA**	**1.434 (0.27) ^a^**	**0.921 (0.16) ^b^**	**1.038 (0.12) ^b^**	**0.984 (0.22) ^a^**	**0.737 (0.09) ^b^**	**0.787 (0.14) ^ab^**
14:0 Mytistic	0.030 (0.01) ^a^	0.016 (0) ^b^	0.025 (0) ^ab^	0.016 (0) ^a^	0.013 (0) ^a^	0.016 (0) ^a^
16:0 Palmitic	1.104 (0.24) ^a^	0.640 (0.11) ^b^	0.747 (0.10) ^b^	0.658 (0.18) ^a^	0.433 (0.06) ^b^	0.541 (0.10) ^ab^
18:0 Stearic	0.294 (0.03) ^a^	0.261 (0.04) ^a^	0.261 (0.01) ^a^	0.300 (0.03) ^a^	0.270 (0.03) ^a^	0.220 (0.03) ^b^
20:0 Arachidic	0.001 (0) ^a^	0 (0) ^b^	0.001 (0) ^ab^	0.002 (0) ^a^	0 (0) ^a^	0 (0) ^a^
**MUFA**	**1.619 (0.29) ^a^**	**0.815 (0.23) ^b^**	**0.615 (0.12)^b^**	**0.582 (0.21) ^a^**	**0.372 (0.05) ^b^**	**0.389 (0.09) ^b^**
16:1 Palmitoleic	0.187 (0.04) ^a^	0.108 (0.03) ^b^	0.132 (0.04) ^b^	0.061 (0.01) ^ab^	0.046 (0.01) ^b^	0.079 (0.02) ^a^
18:1 Vaccenic	0.003 (0) ^a^	0 (0) ^b^	0.002 (0) ^b^	0.008 (0.01) ^a^	0.004 (0.01) ^a^	0.002 (0) ^a^
18:1 Oleic (n-9)	1.315 (0.24) ^a^	0.659 (0.18) ^b^	0.432 (0.06) ^b^	0.455 (0.18) ^a^	0.284 (0.03) ^b^	0.267 (0.06) ^b^
**Total lipids**	**4.873 (0.82) ^a^**	**3.020 (0.60) ^b^**	**2.418 (0.37) ^b^**	**3.023 (0.82) ^a^**	**2.216 (0.28) ^b^**	**2.116 (0.41) ^b^**

FAs content was analyzed by GC performed on liver samples of 9-week-old mice and on serum samples of 6-week-old mice from all groups. Values are expressed as means ± (SD) of *n* = 8 mice/group (mg/100 mg liver or as mg/mL serum). Different superscript letters indicate significant difference (*p* < 0.05) which was determined using a one-way ANOVA followed by Tukey’s test. FAs—Fatty acids, ALA—alpha linolenic acid, EPA—eicosapentaenoic acid, DHA—docosahexaenoic acid, LA—linoleic acid, AA–arachidonic acid, SFA—saturated fatty acid, MUFA—mono-unsaturated acid.

**Table 4 nutrients-12-03494-t004:** Comparison between FA profiles of liver and diets.

	**Control**	
	$\%\frac{\mathrm{FA}\;\mathrm{in}\;\mathrm{the}\;\mathrm{diet}}{\mathrm{PUFAs}\;\mathrm{in}\;\mathrm{the}\;\mathrm{diet}}$	$\%\frac{\mathrm{FA}\;\mathrm{in}\;\mathrm{the}\;\mathrm{liver}}{\mathrm{PUFAs}\;\mathrm{in}\;\mathrm{the}\;\mathrm{liver}}$
18:3 ALA (n-3)	1.70	0.60
20:5 EPA (n-3)	0.05	0.11
22:6 DHA (n-3)	0.37	4.55
18:2 LA (n-6)	97.45	70.47
20:4 AA (n-6)	0.00	18.89
	**Flaxseed**	
	$\%\frac{\mathrm{FA}\;\mathrm{in}\;\mathrm{the}\;\mathrm{diet}}{\mathrm{PUFAs}\;\mathrm{in}\;\mathrm{the}\;\mathrm{diet}}$	$\%\frac{\mathrm{FA}\;\mathrm{in}\;\mathrm{the}\;\mathrm{liver}}{\mathrm{PUFAs}\;\mathrm{in}\;\mathrm{the}\;\mathrm{liver}}$
18:3 ALA (n-3)	76.55	41.89
20:5 EPA (n-3)	0.04	12.32
22:6 DHA (n-3)	0.09	10.25
18:2 LA (n-6)	22.76	29.20
20:4 AA (n-6)	0.01	4.63
	**Fish**	
	$\%\frac{\mathrm{FA}\;\mathrm{in}\;\mathrm{the}\;\mathrm{diet}}{\mathrm{PUFAs}\;\mathrm{in}\;\mathrm{the}\;\mathrm{diet}}$	$\%\frac{\mathrm{FA}\;\mathrm{in}\;\mathrm{the}\;\mathrm{liver}}{\mathrm{PUFAs}\;\mathrm{in}\;\mathrm{the}\;\mathrm{liver}}$
18:3 ALA (n-3)	3.07	1.98
20:5 EPA (n-3)	56.44	20.65
22:6 DHA (n-3)	25.7	41.68
18:2 LA (n-6)	8.61	23.16
20:4 AA (n-6)	3.48	11.40

FA contents in diet and liver samples of 9-week-old mice in all the groups were analyzed by GC. Values are presented as % of total PUFAs in the diet or liver for ALA, EPA, DHA, LA, and AA. FA—Fatty acid, PUFA—poly-unsaturated acid, ALA—alpha linolenic acid, EPA—eicosapentaenoic acid, DHA—docosahexaenoic acid, LA—linoleic acid, AA—arachidonic acid.

**Table 5 nutrients-12-03494-t005:** Measurement of GP width and number of cells.

	**6 Weeks Old**			**9 Weeks Old**		
**Width (µm)**	**Control**	**Flaxseed**	**Fish**	**Control**	**Flaxseed**	**Fish**
Total	119.9 (11.8) ^b^	134.0 (19) ^a^	128.2 (19.5) ^a^	89.8 (9.4) ^b^	101.0(7.4) ^a^	101.1 (10.6) ^a^
PZ	56.3 (9.9) ^b^	65.1 (13.1) ^a^	60.1 (13.8) ^ab^	44.4 (7.9) ^b^	51.8 (6.6) ^a^	51.2 (7.7) ^a^
HZ	63.6 (9.3) ^a^	68.9 (14.2) ^a^	68.0 (15.9) ^a^	45.4 (7.6) ^b^	49.2 (5.9) ^a^	49.9 (8.1) ^a^
**No. of cells**	**Control**	**Flaxseed**	**Fish**	**Control**	**Flaxseed**	**Fish**
Total	12.2 (0.4) ^a^	11.9 (0.7) ^a^	12.2 (1.2) ^a^	10.4 (0.8) ^a^	10.3 (0.7) ^a^	10.4 (0.5) ^a^
PZ	8.6 (0.6) ^a^	7.9 (0.6) ^a^	8.3 (0.6) ^a^	7.1 (0.6) ^a^	6.6 (0.7) ^a^	6.9 (0.1) ^a^
HZ	3.6 (0.2) ^a^	4.0 (0.5) ^a^	3.9 (0.8) ^a^	3.3 (0.3) ^a^	3.6 (0.2) ^a^	3.5 (0.5) ^a^

PZ (proliferative zone); HZ (hypertrophic zone). Values are expressed as means ± SD of *n* = 5 mice/group; different superscript letters indicate significant difference (*p* < 0.05) which was determined using a one-way ANOVA followed by Tukey’s test.

**Table 6 nutrients-12-03494-t006:** Mechanic and morphometric characteristics of the femora in 6- and 9-week-old mice.

	**6 Weeks Old**			**9 Weeks Old**		
**Bone Mechanical Properties**	**Control**	**Flaxseed**	**Fish**	**Control**	**Flaxseed**	**Fish**
Stiffness (N/µ)	0.031 (0.005) ^b^	0.038 (0.002) ^ab^	0.042 (0.009) ^a^	0.063 (0.013)	0.062 (0.011)	0.052 (0.01)
Max load (N)	9.025 (0.82) ^b^	11.264 (1.566) ^a^	12.022 (1.137) ^a^	15.474 (1.826)	15.082 (0.963)	14.121 (2.905)
Yield point (N)	5.07 (0.698)	6.092 (1.537)	7.228 (2.426)	6.628 (1.668)	7.05 (1.657)	6.395 (1.037)
Young’s modulus	0.405 (0.103) ^b^	0.522 (0.118) ^ab^	0.613 (0.178) ^a^	0.994(0.305)	0.895(0.304)	0.72(0.427)
**Trabecular Bone Microarchitecture**	**Control**	**Flaxseed**	**Fish**	**Control**	**Flaxseed**	**Fish**
BV/TV %	8.04 (1.35) ^b^	10.28 (0.8) ^a^	8.81 (1.39) ^ab^	6.02 (1.06) ^b^	8.10 (1.32) ^a^	6.87 (1.82) ^ab^
Tb.Th (mm)	0.03 (0)	0.04 (0)	0.03 (0)	0.04 (0)	0.04 (0)	0.04 (0)
Tb.Sp (mm)	0.22 (0.01)	0.20 (0.02)	0.21 (0.03)	0.27 (0.03)	0.24 (0.03)	0.26 (0.03)
Tb.N (1/mm)	2.21 (0.33) ^b^	2.75 (0.41) ^a^	2.48 (0.46) ^ab^	1.54 (0.25) ^b^	2.00 (0.32) ^a^	1.76 (0.46) ^ab^
**Cortical Bone Microarchitecture**	**Control**	**Flaxseed**	**Fish**	**Control**	**Flaxseed**	**Fish**
B.Ar/T.Ar (%)	30.22 (1.88)	31.67 (1.77)	30.61 (2.79)	33.18 (1.89)	35.66 (2.06)	34.52 (3.19)
M.Ar (mm^2^)	1.39 (0.16)	1.38 (0.11)	1.36 (0.09)	1.36 (0.1)	1.26 (0.1)	1.37 (0.11)
Cs.Th (mm)	0.11 (0.01)	0.12 (0)	0.11 (0.01)	0.12 (0.01)	0.13 (0)	0.13 (0.02)
BMD (gr/cm^3^)	1.17 (0.07)	1.21 (0.06)	1.23 (0.08)	1.33 (0.05) ^b^	1.30 (0.05) ^b^	1.41 (0.06) ^a^

Mechanical properties were evaluated in a three-point bending experiment performed on the bones of all the mice in the three groups. Biomechanical parameters obtained from load–displacement curve: whole-bone stiffness (N/µ); yield point (N); and maximal load (N); to determine the geometric parameters, the bones were scanned by micro-CT. After reconstructing, 2D and 3D analyses were performed. Trabecular bone parameters: bone volume over total volume (BV/TV), trabecular thickness (Tb.Th), trabecular separation (Tb.Sp), and trabecular number (Tb.N). Cortical bone parameters: mean total area (T.Ar), mean bone area (B.Ar), cortical area fraction (Ct.Ar/Tt.Ar), medullary area (Ma.Ar), crossectional thickness (Cs.Th), and bone mineral density (BMD). Values are expressed as means ± (SD) of *n* = 8 mice/group; different superscript letters indicate significant difference (*p* < 0.05) which was determined using a one-way ANOVA followed by Tukey’s test.

**Table 7 nutrients-12-03494-t007:** Five Selected IPA canonical pathways and the genes involved in each pathway.

Comparison	Pathway	Molecules
Flaxseed vs. control	Circadian Rhythm Signaling	Bmal, bhlhe40, cry2, nr1d1, per2, per3, dpb
	Glutathione Redox Reactions	Gstt1, gstt2/gstt2b
	Coagulation System	Plat, serpine1
	Adipogenesis pathway	Bmal1, nr1d2, per2
	Role of Osteoblasts, Osteoclasts and Chondrocytes	Adamts4, adamts5, nfatc2
Fish vs. control	Cell Cycle Control of Replication	Cdc6, cdc7, cdk1, cdk2, cdt1, dbf4, dna2, lig1, mcm2, mcm3, mcm4, mcm5, mcm6, mcm7, orc1, orc2, orc6, pcna, pold1, pole, prim1, top2a
	Role of BRCA1 in DNA Damage Response	Bard1, blm, brca1, brca2, cdkn1a, e2f1, e2f2, e2f4, e2f7, e2f8, fancd2, fancl, gadd45a, mdc1, plk1, rad51, rb1, rbl1, rfc3, rfc4, topbp1
	Heme Biosynthesis II	Alad, cpox, fech, hmbs, ppox, urod, uros
	Circadian Rhythm Signaling	Bmal1, cry2, nr1d1, per2, per3
	Hedgehog Signaling	Ccnb1, cdk1, gli2, prkar2b
Flaxseed vs. Fish	Heme Biosynthesis II	Alad, cpox, fech, ppox, urod, uros
	Role of BRCA1 in DNA Damage Response	Brca1, brca2, e2f2, e2f4, fancd2, topbp1
	Cell Cycle Control of Replication	Mcm3, mcm5, orc1, pole
	Cytokine Signaling	Blvrb, ccr5, il1rl1, socs3
	Hypoxia Signaling	Ube2c, ube2l6, ube2o, ube2t

Differential expressed cutoff was set at *p*-value ≤ 0.05, the log-2-fold change was ≥0.585, min counts was ≥30. Raw *p*-values were adjusted for multiple testing using Benjamini and Hochberg’s procedure. Data were analyzed using IPA (QIAGEN Inc.).

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
