# Peer review of "The Role of Omega-3 Polyunsaturated Fatty Acids from Different Sources in Bone Development"

_nutrients, 2020, doi:10.3390/nu12113494_

Round 1
Reviewer 1 Report
The manuscript by Rozner et. al. entitled, "The Role of Omega-3 Polyunsaturated Fatty Acids from Different Sources in Bone Development" sought to understand how dietary omega-3 polyunsaturated fatty acids affect skeletal development in mice. The authors performed a standard feeding experiment where mice were maintained on diets that differ in fat source, and later assayed for a number of bone phenotypes as well as transcriptome signatures.
It appeared the authors had some supplementary materials, but I cannot find them from the review portal.
The experimental setup was a little confusing. It seemed the authors had a 2-factor design, one being the different diet (control/flaxseed/fish), the other being the different treatment period (3 weeks vs 6 weeks). However, the authors only mentioned that the mice were randomly divided into three dietary groups and did not specify how the 3 weeks group vs 6 weeks group was determined. Moreover, it appeared that the 3 week group was in large neglected in the downstream analyses, as the authors for the most part only showed results for the 6 weeks group. I would like to know the authors’ rationale behind this choice.
The authors should clarify how sections were selected for histological analyses (line 133-134).
The authors should clarify whether the RNA-seq analysis was performed in the 3 weeks group or 6 weeks group or pooled.
For the RNA-seq experiment , although extracting RNA from bone can be difficult, an RIN cut-off at 6 is somewhat concerning.
For the dietary fatty acid analysis (section 3.1), the authors should clarify how the food was handled for the different time intervals (e.g. was food stored in freezer throughout the times? or exposed in similar condition to the actual feeding regime?)
Part of Figure 2 legend should go into methods section.
Table 2 is very difficult to read. I recommend the authors highlight/bold those values that are significantly higher.
Table 3 is also very difficult to read. And I was not convinced it “demonstrated the modifications occurred in the liver”, please clarify the logic.
What does the asterisks in Fig.3A represent? I thought it indicates significant difference, but the authors claimed that there were “no significant differences between the three groups (Figure 3A-D)” (line 311-312). Also, shouldn’t there be error bars for Fig.3A too?
The error bars in Fig. 3B does not seem correct. I understand there will be overlaps between the groups, but it seems only one of the groups had error bars.
Please specify how ALP activity was assessed (line 329-330).
Author Response
Thanks for handling our manuscript titled: “TheRoleofOmega-3 PolyunsaturatedFattyAcidsfromDifferentSources inBoneDevelopment” (ManuscriptID: nutrients-962602)
We are happy for the opportunity to revise it and resubmit for further consideration.
The revised manuscript stemming from reviewer’s judgmentis resubmitted here, and as required includes modifications, clarification and new data. These modifications include:
A new figure (Fig 5), modified Figures (Fig 3 & 4), Three modified tables (Table 2,3 & 5).
We believe that the manuscript in its present version is improved, and hope that it is now suitable for publication in Nutrients.
Dear Reviewers, we greatly appreciate the time and effort you took to evaluate our paper and construct a very relevant set of comments. We feel that by addressing the points you raised and by adding explanation and data accordingly, the paper is genuinely improved.
Response to reviewer #1 comments:
It appeared the authors had some supplementary materials, but I cannot find them from the review portal.
Sorry, it was probably a mistake of editing or uploading. In the revised version, we include all the data in the manuscript except for a supplementary table of all genes that were differentially expressed in the RNA-seq, that found in the supplementary material.
The experimental setup was a little confusing. It seemed the authors had a 2-factor design, one being the different diet (control/flaxseed/fish), the other being the different treatment period (3 weeks vs 6 weeks). However, the authors only mentioned that the mice were randomly divided into three dietary groups and did not specify how the 3 weeks group vs 6 weeks group was determined. Moreover, it appeared that the 3-week group was in large neglected in the downstream analyses, as the authors for the most part only showed results for the 6 weeks group. I would like to know the authors’ rationale behind this choice.
The experiment included 6 groups of post-weaning mice fed with 3 different diets (control/ flaxseed oil/ fish oil), for either 3 or 6 weeks’ period of diet consumption. These time points enabled us to evaluate the changes over the period of growth, one time point during the fastest growth period (6-weeks old mice) and one close to its end (9-weeks old mice). The differences between these periods in terms of growth rates are very clear in Figure 3, and although there were almost no differences between the diets in BW and body length, further analyses in higher resolution reveal the effects of the various diets.
Not all parameters were tested in both time points. RNA-seq, for instance, were performed only on samples from 6-weeks old mice, in their fastest growth period. While growth parameters (BW and length) were performed weekly, along all the experiments. Full bone analyses, including micro-CT, mechanical tests and GP histology, were performed at these two-time points.
This information was corrected and more clearly provided in the material and method section (lines 83-86). We now included also the GP histology in both ages and provided their anthropometric measurements in a modified figure (Figure 4).
The authors should clarify how sections were selected for histological analyses (line 133-134).
Longitudinal median sections from the proximal tibia growth plate were subjected to histological staining and measurements. Measurements were performed on these sections from 5 different animals in each group. In each slide 10 randomly locations throughout the GPs were selected and measured. This is explained in the Material and Method section (lines 134-139).
The authors should clarify whether the RNA-seq analysis was performed in the 3 weeks group or 6 weeks group or pooled.
The RNA-seq was performed on bones from 6-weeks old mice, after 3 weeks of diet consumption. There was no pooling of samples. Each RNA was extracted separately from the isolated bone of each mouse and then subjected to cDNA library preparation and sequencing. The statistic was performed on all the gene expressed by 6 cDNA samples from each group. All this is explained in the Material and Method section (line 177) and in short in the Results (line 408).
For the RNA-seq experiment , although extracting RNA from bone can be difficult, an RIN cut-off at 6 is somewhat concerning.
Yes, the recommended RIN is 8, however several studies have used lower qualitative RNA samples (usually RIN of 6-8) in cases of clinical limitations or challenging samples. We have used the facilities and services of the Grand Israel National Center for Personalized Medicine (G-INCPM), Weizmann Institute of Science, Israel, for both the sequencing and the bioinformatics analyses. The G-INCPM is the most prestige in this field due to its acquired experience with various experimental sets and clinical studies. Accordingly, the number of samples (separate biological repeats) and the used method for libraries and sequencing, as well the quality control steps during library preparation and sequencing were customized. The Illumina TruSeq stranded total RNA protocol permits the use of partially degraded RNA as starting material by modifying the protocol according to the RNA integrity. For instance, the incubation time for total RNA fragmentation was modified according to the RIN. The sequencing was performed in two separate lanes in order to get higher reads per sample and the quality control proves to quality of the samples analyzed. The q-score of all samples were ~36, Q>30 is considered a benchmark for quality in next-generation sequencing. Additionally, validation methods and filtering in the bioinformatics analyses enable the production of qualitative results.
The methods used for sequencing and bioinformatics are detailed in Material and Methods (lines 183-191) with the relevant references.
https://www.agilent.com/cs/library/applications/application-alacris-monitoring-library-prep-omics-4150-tapestation-5994-0946en-agilent.pdf
For the dietary fatty acid analysis (section 3.1), the authors should clarify how the food was handled for the different time intervals (e.g. was food stored in freezer throughout the times? or exposed in similar condition to the actual feeding regime?)
The food for this analyses were thawed and then stored at room temperature for 72 hours, samples were tested in the indicted times. This was done in order to validate the FA profile consumed by the mice, since the food in the cage was measured and replaced twice a week. Consequently, these measurements are equal to the feeding regime during the experiment. This explanation was added in the Results (lines 239-241).
Part of Figure 2 legend should go into methods section.We changed it, Figure 2 legend was shortened, and moved to Material and Methods (lines 110-115).
Table 2 is very difficult to read. I recommend the authors highlight/bold those values that are significantly higher. The major values in the table were modified to Bold.
Table 3 is also very difficult to read. And I was not convinced it “demonstrated the modifications occurred in the liver”, please clarify the logic.
We agree that this table was to loaded, thus it was modified to include only the major PUFA that were affected by the different sources of omega 3. The results of the arachidonic acid that differ between the diets and reveal the shift in enzymatic activity due to substrate abundance were highlighted.
What does the asterisks in Fig.3A represent? I thought it indicates significant difference, but the authors claimed that there were “no significant differences between the three groups (Figure 3A-D)” (line 311-312). Also, shouldn’t there be error bars for Fig.3A too? The error bars in Fig. 3B does not seem correct. I understand there will be overlaps between the groups, but it seems only one of the groups had error bars.
Figure 3 was modified to include all error bars in all time points in 3A and 3B. They indeed are overlapping, but since this is the case, no differences in growth pattern between the groups fed with the different diet, we agree that this presentation is more correct. The asterisk in the figure showed 2 time points in which differences in BW were significant, however since this was not continuous and in general growth pattern was not affected, we claim that there were no differences between the group.
Please specify how ALP activity was assessed (line 329-330).
The full description of ALP staining is described in materials and methods (lines 126-130). Short explanation regarding the principle of ALP staining was added to results section (line 334). Alkaline phosphatase (ALP) staining based on colorimetric insoluble substrate of the enzyme, thus indicating ALP activity in-situ.
Reviewer 2 Report
Reut et al. demonstrated that C57BL6 mice fed with flaxseed oil or fish oil diet decreased fat accumulation in the liver, alter fatty acid content in liver and serum, improve bone mechanical properties and trabecular bone’s microarchitecture. It is an interesting finding, but it requires major improvements.
Major critiques:
- On page 4 line 171, The authors removed bone marrow cells from ulna and humerus bones and only used hollowed bone shafts to extract RNA for gene analysis. This is a big concern, since bone marrow contains both bone marrow stromal cells and monocytes, which have significant impact on osteogenesis and osteoclastogenesis. The author should include the bone marrow cells for RNA extraction and bone gene transcription analysis.
- In Figures (2, 3, 4) and Tables (2, 4). The superscript letter a, b, c, ab need to be clarified. It is unknown what the differences among these superscript letters. Which groups have been compared and the p value for a, b, ab, and c.
- On page 11, line 331. The author mentioned that at the age of 9 weeks, the total GP thickness as well as the thickness of the specific region was significantly higher in both flaxseed the fish groups compared with control mice. From Figure 4A, it looks like the fish group increased thickness of proliferative zone, but flaxseed group had similar thickness of proliferative zone compared with control group. It is better to have detailed number of width of total GP, PZ, and HZ instead of Figure B graph. The author also mentioned supplement Figure 1A on ling 334, which I did not see the supplement Figure 1A.
- On page 12 line 369, the authors mentioned that treatment with flaxseed or fish oil increased BV/TV%, number of trabecular bone. They should show the micro-CT image of femur of mice along with the table 4 data.
- On page 14 line 398 on bone’s transcriptional regulation. This manuscript focus on how omega-3 PUFA regulates bone development. However, the data (shown in Figure 5) mostly are not directly related to osteogenesis and osteoclastogenesis. They mentioned that they analyzed 19,040 genes. It is unclear which genes associated osteogenesis and osteoclastogenesis that they have analyzed. If there were no significant difference among treatment groups. They should list these osteoclastogenic genes and osteogenic genes in the manuscript.
- In the Figure 5 c, the author list ccna2, ccnb1, ccnb2, ccne1, and other genes. However, I cannot find these genes in the Table 5 under cell cycle section.
- In the discussion section, it is hard to understand the relationship of inhibition of cell cycle with the bone development. Figure 5c shows increase of cell cycle canonical pathway. However, the Figure 6 indicates that omega 3 inhibits cell-cycle.
- Previously Beatrice Y. Y. Lau et al. wrote a review (Investigating the Role of Polyunsaturated Fatty Acids in Bone Development Using Animal Models, Molecules 2013, 18, 14203-14227; doi:10.3390/molecules181114203). It looks like previous animal studies had conflict results in the effect of flaxseed oil and fish oil to bone development. Most previous studies used male animals. In this study, you used female mice. Is there a gender difference between male and female? Is there a difference of faxseed oil and fish oil percentage between your study and their studies? You should include discussion of these discrepancy among different studies. It seems like uptake of omega-3 maybe benefit at early growth period in animals. Will uptake omega-3 benefit human bone growth?
Author Response
Dear Reviewers, we greatly appreciate the time and effort you took to evaluate our paper and construct a very relevant set of comments. We feel that by addressing the points you raised and by adding explanation and data accordingly, the paper is genuinely improved.
Major critiques:
On page 4 line 171, The authors removed bone marrow cells from ulna and humerus bones and only used hollowed bone shafts to extract RNA for gene analysis. This is a big concern, since bone marrow contains both bone marrow stromal cells and monocytes, which have significant impact on osteogenesis and osteoclastogenesis. The author should include the bone marrow cells for RNA extraction and bone gene transcription analysis.
We aware of the importance of bone marrow to developmental processes in the long bone. As rightly stated by the reviewer it could be interesting to study the transcriptome profile of the marrow upon the diet consumption. However, in this study we decided to focus on bone cells transcriptome, which is already quite complex due to the mixture of at least 3 types of cells. Osteoblasts, osteocytes and osteoclasts. The focus on these cells and not on cartilage and bone marrow cells derived from the modifications observed in bone phenotype. We do not exclude the possibility that these modifications are affected by the interactions between the adjacent tissues, but mixing even more types of cells in the sequence analyses may blur the specific transcriptional story of the bone. The processes undergoing in the marrow as well as in the growth plate in such nutritional models are of great interest, however they are beyond the scope of this research, that choose to focus on bone tissue and its cells.
In Figures (2, 3, 4) and Tables (2, 4). The superscript letter a, b, c, ab need to be clarified. It is unknown what the differences among these superscript letters. Which groups have been compared and the p value for a, b, ab, and c.
The statistical analyses in these figures and tables are one-way ANOVA followed by Tukey’s test that compared all three groups. The values are expressed as means ± SD of n=8 mice/group, and the different letters indicate significant difference (p< 0.05), thus each group that has different letters, differ significantly from the other groups, groups that share letters do not differ from each other. This is explained in Materials and methods (lines 222-226).
On page 11, line 331. The author mentioned that at the age of 9 weeks, the total GP thickness as well as the thickness of the specific region was significantly higher in both flaxseed the fish groups compared with control mice. From Figure 4A, it looks like the fish group increased thickness of proliferative zone, but flaxseed group had similar thickness of proliferative zone compared with control group. It is better to have detailed number of width of total GP, PZ, and HZ instead of Figure B graph. The author also mentioned supplement Figure 1A on ling 334, which I did not see the supplement Figure 1A.
The Figure was modified to include also the information about the GP at the age of 6 weeks and the detailed numbers of GP width and number of cells are presented in table instead of graphs as requested (new Figure 4), in order to be more precise in the presentation of these results.
On page 12 line 369, the authors mentioned that treatment with flaxseed or fish oil increased BV/TV%, number of trabecular bone. They should show the micro-CT image of femur of mice along with the table 4 data.
Micro-CT representative pictures of the bones were added as a new figure (Figure 5)
On page 14 line 398 on bone’s transcriptional regulation. This manuscript focus on how omega-3 PUFA regulates bone development. However, the data (shown in Figure 5) mostly are not directly related to osteogenesis and osteoclastogenesis. They mentioned that they analyzed 19,040 genes. It is unclear which genes associated osteogenesis and osteoclastogenesis that they have analyzed. If there were no significant difference among treatment groups. They should list these osteoclastogenic genes and osteogenic genes in the manuscript.
Indeed, most of the differential expressed genes are not directly related to osteogenesis and osteoclastogenesis, still a few bone matrix and mineralization related genes were among the modified genes (Figure 6D and supplementary table). For instance, matrix protein collagen type 5α3 expression was up-regulated in the flaxseed oil group compared with the control group, the expression of collagen type 27α1 and 24α1 was down-regulated in the fish group compared with the control group. The comparison of Matrix Gla protein expression between the flaxseed oil and fish oil groups showed an up-regulation in the former. The matrix metalloproteinases ADAMTS-4, 5, and 15 were up-regulated in both the flaxseed oil and fish oil groups compared with the control group. Osteoclastogenic markers, calcitonin receptor, and the nuclear factor of activated T-cells, cytoplasmic 2, were significantly down-regulated in the fish group and flaxseed group compared with the control group. The mild modifications are not surprising since bone phenotypes although differed between groups were not extreme. The differentially expressed genes were related to regulatory pathways of the cells as demonstrated in the figures and text.
We checked also for the expression levels of bone related genes in order to validate the data. The findings in the transcriptome analysis were as expected: the most highly expressed genes (>100,000 counts per sample) were collagen encoding genes, which are the major bone matrix protein expressed by osteoblasts (COL1a, COL1b). Osteopontin (Spp1), Cathepsin K (Ctsk), Osteocalcin (Bglap), TRAP (Apc5), bone sialoprotein (Ibsp), and alkaline phosphatase (Alpl) are matrix proteins produced by osteoblasts and osteoclasts and were also highly expressed(>1000 counts per sample). Osteocytes typical genes were also indicated in this analysis, including ORP150, CapG, DMP1 and PDPN. This information is found in the results (lines 452-465) and discussion (line 565-571).
In the Figure 5 c, the author list ccna2, ccnb1, ccnb2, ccne1, and other genes. However, I cannot find these genes in the Table 5 under cell cycle section.
The genes in table 5 were taken from the IPA analysis tool (QIAGEN Inc, https://www.qiagenbioinformatics.com/products/ingenuitypathway-analysis) that according to the differential expressed genes in the sequences created canonical pathways that were mostly affected by the treatments. It does not include of course all the related genes, that can be found in the excel table in the supplement material. Nevertheless, the ccn genes that were differentially expressed are now added to table 5 in the cell cycle pathway.
In the discussion section, it is hard to understand the relationship of inhibition of cell cycle with the bone development. Figure 5c shows increase of cell cycle canonical pathway. However, the Figure 6 indicates that omega 3 inhibits cell-cycle.
Thanks for this comment. Indeed, the major findings are increase in cell cycle related genes by Omega-3. However, these upregulated genes (Figure 6C) are cell cycle inhibitors, which leads to inhibition of cell cycle progression, as described in the discussion and in figure 7. This subject is now better emphasized in results (lines 445-450) and the discussion (lines 664-668).
Previously Beatrice Y. Y. Lau et al. wrote a review (Investigating the Role of Polyunsaturated Fatty Acids in Bone Development Using Animal Models, Molecules 2013, 18, 14203-14227; doi:10.3390/molecules181114203). It looks like previous animal studies had conflict results in the effect of flaxseed oil and fish oil to bone development. Most previous studies used male animals. In this study, you used female mice. Is there a gender difference between male and female? Is there a difference of flaxseed oil and fish oil percentage between your study and their studies? You should include discussion of these discrepancy among different studies. It seems like uptake of omega-3 maybe benefit at early growth period in animals. Will uptake omega-3 benefit human bone growth?
Thanks for drawing our attention to this review. We added it to our references and used it for discussion of interesting points. A discussion regarding gender issues in the studies of omega-3 influences on the skeleton is found in lines 491-494, and the differences in diets used to study the role of dietary omega-3 are found in lines 536-539. Of-course this type of study cannot serve for nutritional recommendations, but rather for understanding the mechanisms of Omega 3 activity in the context of bone health. Nevertheless, a discussion regarding the relevancy of the results to human skeletal health are found in the introduction and in the first paragraph of the discussion with emphasis in lines 487-488.
Round 2
Reviewer 2 Report
Title: The Role of Omega-3 Polyunsaturated Fatty Acids from Different Sources
in Bone Development
Authors: Reut Rozner, Janna Vernikov, Shelley Griess-Fishheimer, Tamar
Travinsky, Svetlana Penn, Betty Schwartz, Ronit Mesilati-Stahy, Nurit
Argov-Argaman, Ron Shahar, Efrat Monsonego Ornan *
Thanks for the authors providing the GP width and micro-CT images in the figures.
- In the Results section line 429 to 434, the authors only described some of the genes in Figure 6C, such as RB1, p18, p107, p21. They did not describe other genes in Figure 6C, such as Ccna2, Ccnb1, Ccnb2, Ccne1, CcNE2, Cdk1, Cdk2. They should describe these genes in the result section and indicate what the functions of these genes (suppress or promote cell cycle).
- In Figure 6D, the author listed IL1rL1 (interleukin 1 receptor 1-like 1). However, the author did not describe this gene in the result section. The author listed this gene under IL-10 signaling in Table 5. This should be a cytokine signaling (a member of the Toll-like receptor superfamily), not specific to IL-10 signaling. The author should describe how this gene is related to bone remodeling process in the result section. Otherwise, if this gene is not specific to bone remodeling, you should delete it from the figure 6D.
- The authors claimed that they have analyzed COL1a, COL1b, Spp1, Ctsk, Bglap, Acp5, Ibsp, Alpl, and other bone associated genes. They should describe these gene expression in the result section. They should also list these genes under Table 5, since this paper mainly focused on fatty acid effects on bone development.
- In the Figure 6E, the authors list Bmal1, Clock, Dbp genes. They should include these genes in Table 5 under Circadian Rhythm Signaling pathway.
- In line 326-334, the authors described increase matrix production and increase GP thickness. The authors should discuss how the gene analysis profile contribute to matrix production.
Author Response
Thanks for handling our manuscript titled: “TheRoleofOmega-3 PolyunsaturatedFattyAcidsfromDifferentSources inBoneDevelopment” (ManuscriptID: nutrients-962602)
Dear reviewer, we appreciate your effort. The revised manuscript based on your commentsis resubmitted here, and includes minor modifications and explanations. These modifications include: modified Figure 6 and modified table 5 as well as text (marked).
We hope that it is now suitable for publication in Nutrients.
Response to reviewer #2 comments:
In the Results section line 429 to 434, the authors only described some of the genes in Figure 6C, such as RB1, p18, p107, p21. They did not describe other genes in Figure 6C, such as Ccna2, Ccnb1, Ccnb2, Ccne1, CcNE2, Cdk1, Cdk2. They should describe these genes in the result section and indicate what the functions of these genes (suppress or promote cell cycle).
The additional genes are described in short in the results section (lines 44-447), and their role in cell cycle is mentioned.
In Figure 6D, the author listed IL1rL1 (interleukin 1 receptor 1-like 1). However, the author did not describe this gene in the result section. The author listed this gene under IL-10 signaling in Table 5. This should be a cytokine signaling (a member of the Toll-like receptor superfamily), not specific to IL-10 signaling. The author should describe how this gene is related to bone remodeling process in the result section. Otherwise, if this gene is not specific to bone remodeling, you should delete it from the figure 6D.
The section IL10 signaling in table 5 was modified to cytokine signaling, and IL1rL1 was removed from the graph in figure 6D in order to make this part more comprehended. We agree that this gene is not of strong connection to bone remodeling, thus the information is found in the supplemental table (with all other differentially expressed gens) and the results section is focusing on the more significant genes.
The authors claimed that they have analyzed COL1a, COL1b, Spp1, Ctsk, Bglap, Acp5, Ibsp, Alpl, and other bone associated genes. They should describe these gene expression in the result section. They should also list these genes under Table 5, since this paper mainly focused on fatty acid effects on bone development.
These genes are not belonging to table 5 which focus on the differentially expressed genes that were clustered to established pathways by IPA pathway enrichment analysis. We mentioned these genes and their expression levels in the discussion (lines 567-573) as validation tool to the purification technique and the RNA-Seq quality, not as DE genes.
In the Figure 6E, the authors list Bmal1, Clock, Dbp genes. They should include these genes in Table 5 under Circadian Rhythm Signaling pathway.
Thanks for the notification, this error was corrected in table 5.
In line 326-334, the authors described increase matrix production and increase GP thickness. The authors should discuss how the gene analysis profile contribute to matrix production.
The histological analyses were performed on the growth-plate and the adjacent metaphyseal bone tissue, and are mostly relevant to the cells in the epiphyses and the chondro-osseous junction. While for the RNA extraction and RNA-seq analyses, the distal and proximal epiphyses were excised and the diaphyseal bone marrow was removed by centrifugation. Accordingly, the RNA-seq results does not include the chondrocyte seen in the histological section, but more relevant to the micro-CT analyses (Lines 176-181).
Indeed, the findings in the transcriptome analysis are as expected key genes of bone, including typical genes for osteoblast, osteoclasts and osteocytes. In contrast to the expression levels of typical chondrocytic genes that were very low, as expected. Thus we cannot conclude from the gene analysis profile in the bone tissue on the processes that contribute to matrix production in the cartilaginous tissue.
This manuscript is a resubmission of an earlier submission. The following is a list of the peer review reports and author responses from that submission.
Round 1
Reviewer 1 Report
In the submitted manuscript, titled “The Role of Omega-3 Polyunsaturated Fatty Acids from Different Sources in Bone Development,” Rozner et al. reported that the dietary intake of n-3 PUFA altered fat accumulation and fatty acid levels in the liver and serum. Moreover, n-3 PUFA intake improved the mass, microarchitecture, and mechanical properties of bone. Interestingly, transcriptome analysis by RNA-seq showed fluctuations in the expression of genes, such as those related to circadian rhythm and cell cycle, in bone following the intake of n-3 PUFA-rich diet. However, I have several concerns regarding the methodology, including the experimental design and data evaluation, as listed below.
- Although fat accumulation and fatty acid levels changed in the liver and serum, the authors did not mention anything about bone marrow fat accumulation, which is known to increase the fragility of bones.
- As revealed by the histological analysis, total GP thickness was higher in both flaxseed and fish groups than that in the control group. The reasons behind this observation should be discussed and a detailed analysis should be conducted.
- The results obtained by RNA-seq should be validated in vitro and/or in vivo.
- Although the authors focused on osteoclasts and osteoblasts, based on RNA-seq results, no change in TRAP+ and ALP+ cells between the groups was observed via histological analysis. Thus, it is unclear as to which cell type is responsible for the phenotype. Moreover, because whole bone samples were used for analysis, osteocytes should have been included.
- The authors have described the VDR-mediated pathway in the Discussion section and in Figure 6. Was there any increase in the levels of 1,25(OH)2D3 or VDR expression in the samples?
Author Response
Dear Reviewers, we greatly appreciate the time and effort you took to evaluate our paper and construct a very relevant set of comments. We feel that by addressing the points you raised and by adding explanation and data accordingly, the paper is genuinely improved.
Response to reviewer #1 comments:
Although fat accumulation and fatty acid levels changed in the liver and serum, the authors did not mention anything about bone marrow fat accumulation, which is known to increase the fragility of bones.
We thank the reviewer for this comment, we agree that this subject is of interest. Osteoblasts and adipocytes derive from a shared pool of bone marrow mesenchymal stem cells, while metabolic microenvironment, such as nutrition, regulates bone marrow progenitor cells differentiation. The investigation of bone marrow adiposity has increased due to its association with a range of illnesses (osteoporosis, diabetes, anorexia). And as you suggested, several studies showed a tradeoff between bone and fat mass, with the greater differentiation of adipocytes at the expense of osteoblasts thereby leading to reduced bone mass.
In this study, mice bones were used for micro-CT analysis, 3-point bending test and RNA sequencing. Additionally, the amount of bone marrow extracted from bones was too low for GC analysis. We therefore chose to evaluate marrow fat accumulation using tibial histological sections, that can contribute to the understanding of bone marrow composition. Although the adipocyte lipid vacuoles are empty as a result of the ethanol-based dehydration of samples embedded in paraffin blocks, the mature adipocyte ghosts are easily identifiable in staining procedures. Thus, we conducted a double blind visualize evaluation by eight independent examiners on six different slides from each group. Results, when calculated and quantified, show no significant differences between the treatments. Thus, despite the impression of higher level of fat globules in the control group samples as compared to the n-3 enriched groups (as can be seen in the attached figure), we could not establish this fact. We assume that upon higher levels of fat in the diets these differences could become more pronounced. However, this will be part of a future research aimed at this connection that will include higher fat levels in the diet and specific methods (GC, Ossmium, etc) to tackle the topic. A short remark to this issue was added in the discussion about bone mechanical properties (4.2, line 539-548).
Legend: Bone marrow adipocytes in the tibia of 9 weeks-old mice from the 3 different groups.
As revealed by the histological analysis, total GP thickness was higher in both flaxseed and fish groups than that in the control group. The reasons behind this observation should be discussed and a detailed analysis should be conducted.
Indeed, the GP thickness was higher in the groups consuming higher levels of n-3, with no indicated differences in the number of cells in each zone of the GP. These results imply that diet rich in ALA or in EPA and DHA does not affect chondrocytes proliferation, but enhances matrix production, which lead to a thicker GP. The width of the GP was not translated into final increase in bone length, suggesting that dietary n-3 PUFAs in a normal diet have only modest structural effect on the GP. Some support for these results could be found in an in-vitro study that showed that treating chondrocytes with n-3 PUFAs resulted in an increase in cells differentiation and matrix production.
This topic was mentioned in the results section (3.2, line 336-342) and the detailed analyses were explained in M&M 2.5 and Results Fig 4 and text, supplement Fig 1 and legend.
The results obtained by RNA-seq should be validated in vitro and/or in vivo.
The use of RNA-seq outcomes aimed on a global picture of the transcriptional modifications occur in the bone following consumption of different dietary PUFAs.
Schurch et al. study provide guidelines for RNA-seq study design. They found that DESeq2 tool provided a superior true positive identification rate and well-controlled false discovery rate at lower fold changes; DESeq2 tool was used in the bioinformatics analysis. Furthermore, six biological replicates per condition were used in order to ensure valid biological interpretation of the results, as recommended by the author. We believe that these provide solid validation for the reliability of our data [1].
Previous studies have shown close correlations between qPCR and RNA-seq data [2][3][4]. Ideally, re-validation should be done in a separate cohort of samples, and various methods rather than by qPCR. We addressed the reproducibility of our findings, by extensive comparison of the findings with published data (both in-vivo and in-vitro), and by alignment of bone's phenotype with the gene expression data obtained from the RNA-seq.
Although the authors focused on osteoclasts and osteoblasts, based on RNA-seq results, no change in TRAP+ and ALP+ cells between the groups was observed via histological analysis. Thus, it is unclear as to which cell type is responsible for the phenotype. Moreover, because whole bone samples were used for analysis, osteocytes should have been included.
Thanks for this comment that identify a possible confusion in the paper. We studied both the area of the GP and the bone in order to give a full description about the effect of n-3 on the bone. Thus the histological analyses (in figures 4 and S1) were performed on the GP and the adjacent metaphyseal bone tissue, and are mostly relevant to the cells in the chondro-osseous junction. While for the RNA extraction and RNA-seq analyses, the distal and proximal epiphyses were excised and the diaphyseal bone marrow was removed by centrifugation. Accordingly, the RNA-seq results does not include the chondrocyte seen in the histological section, but more relevant to the micro-CT analyses.
Indeed, the findings in the transcriptome analysis are as expected key genes of bone; the most highly expressed genes (>100,000 counts per sample) include collagen encoding genes that are highly expressed in osteoblast and found in bone matrix (COL1a, COL1b). Osteopontin (Spp1), Cathepsin K (Ctsk), Osteocalcin (Bglap), TRAP (Apc5), bone sialoprotein (Ibsp) and alkaline phosphatase (Alpl) are major matrix proteins produced by osteoblasts and osteoclasts, and were highly expressed in RNA-seq results (>1000 counts per sample). Osteocytes typical genes were also indicated in this analysis, including ORP150, CapG, DMP1 and PDPN. This is in contrast to a lower expression (less than 150 counts per sample) of collagen type-10 (COL10) typical for hypertrophic zone of the GP that was excluded from the RNA extraction.
To clarify this issue we added the information regarding osteocytes to the discussion (4.3, line 560).
The authors have described the VDR-mediated pathway in the Discussion section and in Figure 6. Was there any increase in the levels of 1,25(OH)2D3 or VDR expression in the samples?
We did not indicate a significant alteration in VDR expression levels between the groups. VDR expression was 393, 435, 383 (normalized counts per sample) in the control, flaxseed oil and fish oil groups, respectively. Furthermore, the serum levels of 1,25(OH)2D3 were not tested, however it is important to note that all diets included the recommended amounts of vitamins and minerals according to AIN-93.VDR is mentioned in the suggested model in Figure 6, since we found many hints in our sequencing data and in the literature to a possible connection between vitamin D and n-3 PUFAs to skeletal health. A remarkable trend within the n-3 PUFAs diet fed groups was the alteration in the core circadian clock and cell cycle genes compared with the control group. Recent studies have also highlighted a putative connection between circadian clock proteins and cell cycle progression [5][6][7], our results demonstrate a relation between these two pathways and their influence on bone development, as affected by dietary n-3 PUFA.
VDR is a ligand-activated nuclear transcription factor that is instrumental for bone health. It is known to affect proliferation and differentiation of osteoblasts and osteoclasts which are crucial for the balance of bone remodeling [8]. Knock-out mice models for the cell cycles proteins pl07-/-; pl30-/- in developing bones reveal severely shortened limbs as a result of disrupted endochondral ossification process [9]. Verlinden et al., found that p107 and p130, are essential mediators of the anti-proliferative activity of 1,25(OH)D3 in mouse osteoblasts[10].
The VITamin D and OmegA-3 TriaL (VITAL), is an ongoing clinical research in over 25,000 men and women across the United-states. The main goal of VITAL is to determine whether vitamin D and/or n-3 PUFAs can prevent cancer, heart disease, and stroke. However, other health outcomes, such as risk for bone fractures, are now being examined as they might have potential therapeutic effect on bones [11]. Interestingly, 1,25(OH)D3 activity in osteoblasts have been recently reported to be under circadian rhythm regulation [12]. Since the circadian clock cluster of genes was also differentially expressed in our experiment, we suggested a role for n-3 PUFAs in circadian rhythm regulation, its possible connection to VDR signaling and their link to the observed skeletal phenotype.
These ideas are better explained and discussed in the discussion part of Figure 6 lines 613-622.

Reviewer 2 Report
This article describes evaluation of the effect of dietary n-3 PUFAs on bone development. The authors showed the role of n-3PUFAs in regulating bone quality and RNA-seq analysis suggested that bone circadian rhythm is involved in its effect. The following points should be considered.
The authors described that consumption of n-3PUFAs affect the thickness of growth plate of total and specific regions in histological evaluation. Whereas longitudinal bone growth unchanged. Please explain how to consider about the point.
The reviewer cannot follow the description on the discussion section in lines 499 to 506. From the description in the Method section, the reviewer understand that the authors extracted total lipids followed the conversion to fatty acid ethyl ester form for GC analysis. But, it only measures fatty acid composition. Did the authors measure FFA levels? Please explain that and provide some interpretation on the role of FFAR4 in osteocytes associated with the bone growth.
The authors consider dietary fish oil decrease number of active osteoclasts associated with the down-regulation of Calcr gene expression. However, the results from histological evaluation showed no difference in TRAP positive osteoclasts. Please explain how to consider about the point.
The provided results on GC analysis is difficult to follow. The reviewer think that individual fatty acid content in the diet, liver, and serum lipid should be provided.
Throughout the manuscript, the authors should use the term “amount” and “percentage” of fatty acid. For example, in the line 286, “% of LA” in PUFA is 2.6 times higher, not “amount”.
In the paragraph in line 247 to 255, “ -fold lower” is not general and is difficult to understand. I think it is better to change the expression.
Author Response
Dear Reviewers, we greatly appreciate the time and effort you took to evaluate our paper and construct a very relevant set of comments. We feel that by addressing the points you raised and by adding explanation and data accordingly, the paper is genuinely improved.
Response to reviewer #2 comments:
The authors described that consumption of n-3 PUFAs affect the thickness of growth plate of total and specific regions in histological evaluation. Whereas longitudinal bone growth unchanged. Please explain how to consider about the point.
This was questioned also by reviewer #1; Indeed, microscopic measurements of the GP dimensions reveal wider plate in mice consumed Flaxseed oil or Fish oil diets compared to control mice. Differences in GP width were not reflected in the femoral length, probably due to different measurement scales; while femoral length was measured on a scale of mm, GP width was measured using microscopic software on a scale of µm that allows detection of slighter changes. The results suggest that dietary n-3 PUFAs as part of a normal diet have only moderate structural effect on the GP. This is also supported by the fact that cell number was not modified by the diet.
Consideration to these results were discussed in results section (3.2, line 336-342).
The reviewer cannot follow the description on the discussion section in lines 499 to 506. From the description in the Method section, the reviewer understand that the authors extracted total lipids followed the conversion to fatty acid ethyl ester form for GC analysis. But, it only measures fatty acid composition. Did the authors measure FFA levels? Please explain that and provide some interpretation on the role of FFAR4 in osteocytes associated with the bone growth.
Thanks for this important request for clarification. The GC analysis was conducted on total lipid extract, isolated from fasting mice.No separation to lipid fractions (triglyceride, phospholipids and FFA) was carried out prior to the extraction procedure. Nevertheless, since plasma was collected under fasting conditions, we assume that the majority of fatty acids were present in serum in their free form, as FFA. This notion was added to M&M (2.1, line 87).
The free fatty acid receptor 4 (FFAR4; also known as GPR120) is expressed in bone cells and preferentially binds n−3 PUFAs. In our study it was expressed in the RNA-seq of all groups without differences. FFAR4 was shown to stimulate bone formation and suppress bone resorption upon activation by n-3 PUFA[13]. We cannot conclude from our results in which bone-cell type the receptor is expressed, osteocytes, osteoblasts, osteoclasts, or all of them. However, FFAR4 pathway is possibly underlying a connection between nutrition, lipid metabolism and bone remodeling and might be crucial to the protective effects of dietary n-3 PUFAs on bones. This was added to the discussion (4.1, line 518-520).
The authors consider dietary fish oil decrease number of active osteoclasts associated with the down-regulation of Calcr gene expression. However, the results from histological evaluation showed no difference in TRAP positive osteoclasts. Please explain how to consider about the point.
Thanks for this comment that identify a possible confusion in the paper. We studied both the area of the GP and the bone in order to give a full description about the effect of n-3 on the bone. Thus, the histological analyses (in figures 4 and 1S) were performed on the GP and the adjacent metaphyseal bone tissue, and are mostly relevant to the cells in the chondro-osseous junction. While for the RNA extraction and RNA-seq analyses, the distal and proximal epiphyses were excised and the diaphyseal bone marrow was removed by centrifugation. Accordingly, the RNA-seq results does not include the chondrocyte seen in the histological section, but more relevant to the micro-CT analyses. Indeed, the findings in the transcriptome analysis are as expected key genes of bone; with high expression levels of genes coding for matrix proteins produced by osteoblasts, osteoclasts and osteocytes (Collagen type I, Osteopontin, Cathepsin K, Osteocalcin, TRAP, Calcitonin receptor, ORP150, CapG, and Dmp1). This is in contrast to a lower expression of collagen type-10 and other genes typical for hypertrophic zone of the GP that was excluded from the RNA extraction. This issue was further clarified indiscussion (4.3, line 555-561) and M&M (2.8).
Throughout the manuscript, the authors should use the term “amount” and “percentage” of fatty acid. For example, in the line 286, “% of LA” in PUFA is 2.6 times higher, not “amount”. This was modified, accordingly.
In the paragraph in line 247 to 255, “ -fold lower” is not general and is difficult to understand. I think it is better to change the expression.This was modified to "times lower".
The provided results on GC analysis is difficult to follow. The reviewer think that individual fatty acid content in the diet, liver, and serumlipid should be provided.
We agree with the reviewer, a table of all FA content in the different samples was added to the Supplementary data.
|
Fatty Acid |
Diet |
Liver |
Serum |
||||||
|
Control (mol%) |
Flaxseed (mol%) |
Fish (mol%) |
Control (mol%) |
Flaxseed (mol%) |
Fish (mol%) |
Control (mol%) |
Flaxseed (mol%) |
Fish (mol%) |
|
|
Total n-3 |
1.13 |
55.47 |
31.10 |
1.73 |
26.37 |
17.67 |
1.98 |
22.10 |
21.67 |
|
18:3 ALA (n-3) |
0.92 |
55.47 |
1.12 |
0.22 |
17.75 |
0.60 |
0.15 |
10.52 |
0.69 |
|
20:5 EPA (n-3) |
0 |
0 |
20.60 |
0.04 |
4.87 |
5.93 |
0.09 |
7.62 |
10.70 |
|
22:6 DHA (n-3) |
0.20 |
0 |
9.38 |
1.47 |
3.74 |
11.13 |
1.82 |
3.95 |
10.27 |
|
Total n-6 |
53.37 |
16.98 |
5.39 |
33.78 |
14.62 |
10.87 |
42.56 |
20.67 |
18.41 |
|
18:2 LA (n-6) |
53.12 |
16.49 |
3.14 |
25.61 |
12.36 |
7.12 |
26.37 |
19.83 |
11.33 |
|
20:4 AA (n-6) |
0 |
0 |
1.27 |
0.27 |
0.02 |
0.07 |
15.32 |
4.23 |
6.48 |
|
SFA |
15.62 |
9.85 |
36.08 |
36.91 |
40.74 |
56.63 |
34.90 |
35.02 |
40.01 |
|
14:0 Mytistic |
0.07 |
0.06 |
11.08 |
0.74 |
0.66 |
1.25 |
0.66 |
0.74 |
0.98 |
|
16:0 Palmitic |
13.59 |
6.22 |
19.97 |
24.27 |
23.01 |
33.61 |
23.74 |
21.35 |
28.14 |
|
18:0Stearic |
1.66 |
3.38 |
3.72 |
5.96 |
8.52 |
10.78 |
10.13 |
12.03 |
10.41 |
|
20:0 Arachidic |
0.29 |
0.29 |
0.16 |
0.03 |
0.04 |
0.05 |
0.07 |
0.11 |
0 |
|
MUFA |
29.86 |
17.68 |
27.41 |
33.14 |
26.58 |
25.54 |
18.85 |
16.83 |
18.58 |
|
16:1 Palmitoleic |
0.12 |
0.09 |
11.54 |
4.20 |
3.85 |
5.90 |
2.21 |
2.28 |
4.10 |
|
18:1 Vaccenic |
0 |
0 |
4.20 |
2.03 |
1.28 |
1.70 |
1.53 |
1.20 |
1.32 |
|
18:1 Oleic (n-9) |
29.51 |
17.46 |
10.37 |
26.55 |
21.21 |
17.61 |
14.46 |
12.73 |
12.50 |
|
Total lipids |
100 |
100 |
100 |
100 |
100 |
100 |
100 |
100 |
100 |
Cole, C.; Schurch, N.J.; Schofield, P.; Gierlin, M.; Sherstnev, A.; Singh, V.; Wrobel, N.; Gharbi, K.; Simpson, G.G.; Owen-hughes, T.O.M.; et al. How many biological replicates are needed in an RNA-seq experiment and which differential expression tool should you use ? 2016, 839–851, doi:10.1261/rna.053959.115.
- Consortium, S.M. Articles A comprehensive assessment of RNA-seq accuracy , reproducibility and information content by the Sequencing Quality Control Consortium. 2014, 32, doi:10.1038/nbt.2957.
- Wang, Z.; Gerstein, M.; Snyder, M. Nrg2484-1. Nat. Rev. | Genet.2009, VOLUME 10, 57–63.
- Griffith, M.; Griffith, O.L.; Mwenifumbo, J.; Goya, R.; Morrissy, A.S.; Morin, R.D.; Corbett, R.; Tang, M.J.; Hou, Y.C.; Pugh, T.J.; et al. Alternative expression analysis by RNA sequencing. Nat. Methods2010,7, 843–847, doi:10.1038/nmeth.1503.
- Gérard, C.; Goldbeter, A. Entrainment of the mammalian cell cycle by the circadian clock: Modeling two coupled cellular rhythms. PLoS Comput. Biol.2012, 8, doi:10.1371/journal.pcbi.1002516.
- Boucher, H.; Vanneaux, V.; Domet, T.; Parouchev, A. Circadian Clock Genes Modulate Human Bone Marrow Mesenchymal Stem Cell Differentiation , Migration and Cell Cycle. 2016, 1–16, doi:10.1371/journal.pone.0146674.
- Guillaumond, F.; Delaunay, F. The Circadian Clock Component BMAL1 Is a Critical Regulator. 2008, doi:10.1074/jbc.M705576200.
- Imai, Y.; Youn, M.; Inoue, K.; Takada, I.; Kouzmenko, A.; Kato, S. NUCLEAR RECEPTORS IN BONE PHYSIOLOGY AND DISEASES NUCLEAR RECEPTORS. 2020, 481–523, doi:10.1152/physrev.00008.2012.
- Berman, S.D.; Yuan, T.L.; Miller, E.S.; Lee, E.Y.; Caron, A.; Lees, J.A. The Retinoblastoma Protein Tumor Suppressor Is Important for Appropriate Osteoblast Differentiation and Bone Development. 2008, 6, 1440–1452, doi:10.1158/1541-7786.MCR-08-0176.
- Verlinden, L.; Eelen, G.; Hellemont, R. Van; Engelen, K.; Beullens, I.; Camp, M. Van; Marchal, K.; Mathieu, C.; Bouillon, R.; Verstuyf, A. of the checkpoint proteins , Chk1 and Claspin , is mediated by the pocket proteins p107 and p130. 2007, 103, 411–415, doi:10.1016/j.jsbmb.2006.12.080.
- Goldman, A.L.; Donlon, C.M.; Cook, N.R.; Manson, J.E.; Buring, J.E.; Copeland, T.; Yu, C.Y.; Leboff, M.S. VIT amin D and OmegA-3 TriaL ( VITAL ) bone health ancillary study : clinical factors associated with trabecular bone score in women and men. 2018.
- Takarada, T.; Xu, C.; Ochi, H.; Nakazato, R.; Yamada, D.; Nakamura, S.; Kodama, A.; Shimba, S.; Mieda, M.; Fukasawa, K.; et al. JB M R Bone Resorption Is Regulated by Circadian Clock in Osteoblasts., doi:10.1002/jbmr.3053.
- Ahn, S.H.; Park, S.Y.; Baek, J.E.; Lee, S.Y.; Baek, W.Y.; Lee, S.Y.; Lee, Y.S.; Yoo, H.J.; Kim, H.; Lee, S.H.; et al. Free fatty acid receptor 4 (GPR120) stimulates bone formation and suppresses bone resorption in the presence of elevated n-3 fatty acid levels. Endocrinology2016, 157, 2621–2635, doi:10.1210/en.2015-1855.

Round 2
Reviewer 1 Report
The manuscript has been revised well. I recommend that this manuscript be accepted for publication.
Author Response
Thank you for your comments
Reviewer 2 Report
Some points have been improved. However, I think that the manuscript still have some problems as indicated below.
The authors argue the beneficial effects of n-3PUFA on bone development based on the RNA-seq analysis. But significant differences were not observed in osteoclasts and osteoblasts in histological analysis. I don’t think that the present set of data are conclusive enough to draw structural conclusions. The article needs additional experiments to validate RNA-seq data and to explain the factors which are responsible for the bone mechanical quality.
Author Response
Thank you for your comments